# Primaquine in glucose-6-phosphate dehydrogenase deficiency: an adaptive pharmacometric assessment of ascending dose regimens in healthy volunteers

Sasithon Pukrittayakamee[1,2], Podjanee Jittamala[1,2], James A Watson[3,4*], Borimas Hanboonkunupakarn[1,2], Pawanrat Leungsinsiri[1], Kittiyod Poovorawan[1,2], Kesinee Chotivanich[1,2], Germana Bancone[4,5], Cindy S Chu[4,5], Mallika Imwong[6], Nicholas PJ Day[2,3], Walter RJ Taylor[2,3†], Nicholas J White[2,3†]

[1]Clinical Therapeutics Unit, Faculty of Tropical Medicine, Mahidol University, Bangkok, Thailand; [2]Mahidol Oxford Tropical Medicine Research Unit, Faculty of Tropical Medicine, Mahidol University, Bangkok, Thailand; [3]Oxford University Clinical Research Unit, Hospital for Tropical Diseases, Ho Chi Minh, Viet Nam; [4]Centre for Tropical Medicine and Global Health, Nuffield Department of Medicine, University of Oxford, Oxford, United Kingdom; [5]Shoklo Malaria Research Unit, Mae Sot, Thailand; [6]Department of Molecular Tropical Medicine and Genetics, Faculty of Tropical Medicine, Mahidol University, Bangkok, Thailand

*For correspondence:
james@tropmedres.ac

†These authors contributed equally to this work

## Abstract

**Background:** Primaquine is an 8-aminoquinoline antimalarial. It is the only widely available treatment to prevent relapses of *Plasmodium vivax* malaria. The 8-aminoquinolines cause dose-dependent haemolysis in glucose-6-phosphate dehydrogenase deficiency (G6PDd). G6PDd is common in malaria endemic areas but testing is often not available. As a consequence primaquine is underused.

**Methods:** We conducted an adaptive pharmacometric study to characterise the relationship between primaquine dose and haemolysis in G6PDd. The aim was to explore shorter and safer primaquine radical cure regimens compared to the currently recommended 8-weekly regimen (0.75 mg/kg once weekly), potentially obviating the need for G6PD testing. Hemizygous G6PDd healthy adult Thai and Burmese male volunteers were admitted to the Hospital for Tropical Diseases in Bangkok. In Part 1, volunteers were given ascending dose primaquine regimens whereby daily doses were increased from 7.5 mg up to 45 mg over 15–20 days. In Part 2 conducted at least 6 months later, a single primaquine 45 mg dose was given.

**Results:** 24 volunteers were enrolled in Part 1, and 16 in Part 2 (13 participated in both studies). In three volunteers, the ascending dose regimen was stopped because of haemolysis (n=1) and asymptomatic increases in transaminases (n=2; one was hepatitis E positive). Otherwise the ascending regimens were well tolerated with no drug-related serious adverse events. In Part 1, the median haemoglobin concentration decline was 3.7 g/dL (range: 2.1–5.9; relative decline of 26% [range: 15–40%]). Primaquine doses up to 0.87 mg/kg/day were tolerated subsequently without clinically significant further falls in haemoglobin. In Part 2, the median haemoglobin concentration decline was 1.7 g/dL (range 0.9–4.1; relative fall of 12% [range: 7–30% decrease]). The ascending dose primaquine regimens gave seven times more drug but resulted in only double the haemoglobin decline.

**Conclusions:** In patients with Southeast Asian G6PDd variants, full radical cure treatment can be given in under 3 weeks compared with the current 8-week regimen.

**Funding:** Medical Research Council of the United Kingdom (MR/R015252/1) and Wellcome (093956/Z/10/C, 223253/Z/21/Z).
**Clinical trial number:** Thai Clinical Trial Registry: TCTR20170830002 and TCTR20220317004.

## eLife assessment

This manuscript addresses an **important** question, that in countries endemic for *P vivax* the need to administer a primaquine (PQ) course adequate to prevent relapse in G6PD deficient persons poses a real dilemma. On one hand PQ will cause haemolysis; on the other hand, without PQ the chance of relapse is very high. As a result, out of fear of severe haemolysis, PQ has been under-used. This manuscript is **convincing** that regimen (1) can be used successfully to deliver within 3 weeks, under hospital conditions, the dose of PQ required to prevent *P vivax* malaria relapse.

## Introduction

Over the past 70 years primaquine has been the only drug widely available to prevent relapses of *Plasmodium vivax* and *Plasmodium ovale* malaria (radical cure). Primaquine has been given to hundreds of millions of patients in single doses to prevent *P. falciparum* transmission, in 5- to 14-day courses for radical cure of vivax and ovale malarias, and during mass treatments (**Ashley et al., 2014**). The main adverse effect of primaquine, and the other 8-aminoquinoline antimalarials, is dose-dependent oxidant haemolysis in individuals with glucose-6-phosphate dehydrogenase (G6PD) deficiency (**Luzzatto et al., 2020**; **Recht et al., 2014**). As G6PD deficiency is common in malaria endemic regions, primaquine is underused because reliable point-of-care testing for G6PD deficiency is usually not available (**Recht et al., 2018**) and prescribers are naturally reluctant to risk causing potentially severe, life-threatening haemolysis (**Yilma et al., 2023**).

Outside sub-Saharan Africa *P. vivax* is the main cause of malaria in most endemic regions, and in these regions relapse is often the main cause of illness (**Commons et al., 2020**). Relapsing infections cause significant morbidity and, in higher transmission settings, mortality in young children, mediated mainly by chronic anaemia (**Douglas et al., 2014**; **Commons et al., 2019**). Relapses are also an important source of *P. vivax* transmission. The underuse of primaquine therefore contributes to substantial morbidity and mortality and to the failure to control and eliminate vivax malaria in endemic areas.

G6PD deficiency is the most common red blood cell enzyme deficiency of humans (**Luzzatto et al., 2020**). G6PD deficiency occurs mainly in malaria endemic or historically endemic regions (**Howes et al., 2012**). Mutations in the *G6PD* gene on the X chromosome confer reduced enzyme stability. This results in impaired erythrocyte defences against oxidant stresses and thereby increases the risk of haemolysis of older G6PD-depleted erythrocytes. More severe *G6PD* deficiency variants are associated with greater drug-induced haemolysis as a broader erythrocyte age range is G6PD-depleted (**Piomelli et al., 1968**). The substantial haemolytic risk from 8-aminoquinoline antimalarials means that the standard 7- to 14-day radical cure primaquine regimens are contraindicated in G6PD deficiency (**World Health Organization, 2015**).

Seminal clinical investigations were conducted over 50 years ago in adults with the African A-variant of G6PD deficiency. These showed that although oxidant haemolysis was inevitable, it affected predominantly the older erythrocytes. Continued primaquine administration to G6PD-deficient subjects resulted in 'resistance' to the haemolytic effect. The selective haemolysis of the older red cells resulted in a compensatory increase in the number of reticulocytes. Thus, the red cell population became progressively younger and increasingly resistant to oxidant stress, so overall haemolysis decreased and a steady state was reached. This suggested a therapeutic strategy of controlled haemolysis which would limit the degree of anaemia by allowing time for the compensatory erythropoetic response (**Alving et al., 1960**; **Kellermeyer et al., 1962**). This mechanistic understanding provided the rationale for the once weekly primaquine regimen of 0.75 mg base/kg for 8 weeks, currently recommended by the World Health Organization in patients with G6PD deficiency (**World Health Organization, 2015**). However, the safety of this regimen in more severe *G6PD* deficiency variants was never established. A small cohort study of the weekly primaquine regimen in 18 G6PD-deficient (17 had the 871G>A [Viangchan] variant) and 57 G6PD-normal adult vivax malaria patients

in Cambodia suggested that single 45 mg doses may not be safe in the more severe *G6PD* deficiency variants (*Kheng et al., 2015*). A quarter of the G6PD-deficient patients had a >25% fall in haemoglobin (compared to none in the G6PD-normal group) and one patient required a blood transfusion for a symptomatic fall in haemoglobin fall from 10 to 7.5 g/dL.

Over the past 50 years there has been substantial variation in national policies and practices. Some countries (e.g. Iran, Myanmar), which did not have G6PD testing available in endemic areas, have recommended the once weekly primaquine regimen as standard practice for all vivax malaria cases. Other countries have recommended giving the 14-day daily courses of primaquine without testing, although this recommendation is often not followed. To develop an alternative shorter and potentially safer approach to primaquine dosing in G6PD deficiency, we conducted a two-part adaptive pharmacometric study in Thailand with the goal of characterising the dose-response relationship for primaquine-induced haemolysis in healthy G6PD-deficient volunteers.

## Methods
### Trial design
The overall objective of this study was to identify radical cure primaquine regimens that would be safer in G6PD deficiency. This required titration of the daily primaquine doses against haemolysis, balancing the need to provide a radical curative dose of primaquine within a reasonable time-frame whilst minimising the inevitable haemolysis.

This was a two-part study, conducted in the Hospital for Tropical Diseases in Bangkok, of primaquine in hemizygote G6PD-deficient male healthy volunteers. Part 1 evaluated the tolerability, safety, and haematological consequences of ascending doses of primaquine and was adaptive. The primaquine regimen was titrated based on the incremental haemoglobin changes observed in the previous participants, continuous safety evaluation by the investigators, and a set of guiding pre-specified rules. This iterative adaptive approach accumulated information to refine the successive regimens. The primary consideration throughout the trial was participant safety. In Part 2 of the study, after a wash-out period of at least 6 months, a single 45 mg (base equivalent) primaquine dose was given, and the volunteers were monitored as in Part 1.

### Study site and participants
The study took place in the Clinical Therapeutics Unit volunteer ward in the Hospital for Tropical Diseases, Faculty of Tropical Medicine, Mahidol University, Bangkok, Thailand. The recruitment and follow-up periods were from November 2018 to October 2020 (Part 1) and June to September 2022 (Part 2). This coincided with and was disrupted by the COVID-19 pandemic. A COVID-19 mitigation plan was implemented when the local lockdown was lifted.

Healthy male volunteers were recruited if they provided written informed consent, had a G6PD enzyme activity <30% of the population median value determined by a validated quantitative spectrophotometric G6PD assay, a known genotype-confirmed G6PD deficiency variant (according to a previously published method; *Bancone et al., 2019*), and were aged between 18 and 65 years with a screening haemoglobin concentration >11 g/dL. Detailed exclusion criteria are provided in Appendix 1.

### Trial procedures
#### Enrolment and primaquine dosing
The risks and the rationale of the study were detailed to potential volunteers and it was explained that they could withdraw from the study at any time if they wished. Primaquine phosphate (Thailand Government Pharmaceutical Organisation, Bangkok, Thailand) was provided as tablets containing either 5 or 15 mg primaquine base equivalent. A tablet cutter was used to split the tablets (smallest dose increment was 2.5 mg base equivalent). Primaquine was given orally following a standardised light snack and subjects were observed closely for the first 4 hr.

#### Part 1
We recruited in cohorts of five volunteers, with an interval of 2 weeks between cohorts to allow sufficient time to analyse the data and determine the next primaquine regimen. The overall goal was

to increase the daily primaquine doses and cause gradual haemolysis which was offset by concomitant reticulocytosis. This would steadily reduce the age of the red cell population and thereby avoid precipitous symptomatic falls in the haemoglobin concentration (*Alving et al., 1960*; *Kellermeyer et al., 1962*). The cumulative total primaquine dose given needed to be sufficient to provide a radical curative effect in the treatment of vivax malaria (i.e. between 5 and 7 mg base/kg) (*Commons et al., 2024*).

Careful monitoring was done throughout to assess the degree of haemolysis and adjust or stop the dosing, as required. The dose regimens in this exploratory investigation (Part 1) were adapted as follows.

The primaquine regimen given to the first five volunteers consisted of four cycles of 5 days daily primaquine dosing (i.e. total 20 days). The ascending doses were 7.5, 15, 22.5, and 30 mg base equivalent, respectively. This initial dose regimen was chosen on the basis of a mathematical model of primaquine-induced haemolysis in G6PD deficiency using earlier data from vivax malaria patients who had received single 45 mg primaquine doses (*Watson et al., 2017*; *Kheng et al., 2015*). Subjects proceeded to the next higher dose cycle if they satisfied several prospectively defined safety criteria (Appendix 2). In this first round the total primaquine dose was 375 mg base equivalent. The results of the first five volunteers were reviewed and the dosing regimens adjusted. This iterative process of review and adjustment continued thereafter. The once daily dosing in each cycle was increased in increments of 2.5 or 7.5 mg (not adjusted for body weight). Once it became clear this rate of dose increase was generally well tolerated, the number of days per cycle was reduced to 3 or 4 days in subsequent subjects to test regimens of shorter duration with faster dose escalation. Subjects were reviewed clinically before each dose increase.

### Part 2

In Part 2, a single dose of 45 mg primaquine base equivalent was administered, with similar careful monitoring for 1 week.

## Monitoring procedures/evaluations

In both studies, volunteers were observed closely. In Part 1, all volunteers were admitted to the ward for 28 days with a subsequent follow-up visit on day 49. For Part 2, at least 6 months later, volunteers were re-admitted for 24 hr on day 0, reviewed daily until day 7, and then again on day 14.

At enrolment all volunteers underwent a detailed clinical examination. Thereafter, at each assessment volunteers were asked how they felt, whether they had any adverse effects, had their vital signs measured, and two blood samples were taken for measurement of haemoglobin concentration (HemoCue, Ängelholm, Sweden). The average of the two results was recorded. Methaemoglobin (%) was measured non-invasively at least once daily using a Masimo Rad 57 oximeter, and urine colour was recorded twice daily. Haemoglobinuria was assessed visually using a modified Hillmen score (*Hillmen et al., 2004*).

Wright-Giemsa-stained and new methylene blue-stained blood films were prepared for red cell morphology, and manual reticulocyte and Heinz body counts, respectively. In Part 1 these were done daily from day 0 (day of first primaquine dose) until day 20, and then on days 22, 24, 26, and 28, and finally on day 49. In Part 2 they were done at every visit. Other laboratory investigations included a complete blood count (CBC) and reticulocyte count, routine biochemistry (including lactate dehydrogenase [LDH] and haptoglobin), plasma haemoglobin, and plasma primaquine and carboxyprimaquine concentrations. In Part 1 these were done at screening and then every 3–5 days (start of each new cycle, i.e. dosing increment); in Part 2 they were done at screening, and on days 3, 7, and 14.

The G6PD deficiency variants were genotyped as described previously (*Bancone et al., 2019*; *Boonyuen et al., 2021*). Haemoglobin electrophoresis and genotyping of the common cytochrome P450 2D6 (CYP2D6) genotypes found in Southeast Asia was also performed (*Puaprasert et al., 2018*). Presumptive alpha-thalassaemia trait (which is very common in Thailand) was defined as a mean cell volume <80 fl or a mean cell haemoglobin <27 pg, and HbA2 ≤4.4. Additional laboratory measurements were taken if unplanned dose adjustments were necessary, or there was a clinical indication.

## Safety monitoring and stopping rules for Part 1

The pre-specified rules for adjusting primaquine dose regimens across cohorts in Part 1 are illustrated in *Appendix 2—figure 1*, and the rules for increasing primaquine doses for each enrolled subject are shown in *Appendix 2—figure 2*. The overall aim was to titrate dosing in order to obtain small daily falls in haemoglobin of between 0.1 and 0.2 g/dL. For a given subject, primaquine doses were increased only if the haemoglobin concentration was >9 g/dL, was >70% of baseline, the urine Hillmen score was ≤5, and the subject felt well and had no symptoms of anaemia.

We defined haemolysis which would result in stopping primaquine (study withdrawal), as any one of the following:

- >40% fall in haemoglobin from baseline;
- a haemoglobin below 8 g/dL (irrespective of symptoms);
- a fall in haemoglobin associated with clinically significant signs of haemolysis: jaundice, passing dark urine (Hillmen score ≥6), evidence of acute renal injury (≥2-fold increase in serum creatinine from baseline), or hyperkalaemia (serum potassium >5.2 mmol/L).

Any individual whose laboratory tests met these criteria remained in hospital and was monitored closely until resolution of signs and symptoms, and haemoglobin concentrations had reached at least 10 g/dL. Blood transfusion was available at any time if needed if volunteers had a symptomatic fall in haemoglobin to below 8 g/dL.

An Independent Drug Safety Monitoring Board was established to review the data after each cohort had completed their follow-up and gave feedback before the next cohort was allowed to proceed. For both studies, adverse events were recorded and graded using the Common Toxicity Criteria v 5.0 for grading adverse events.

## Sample size

As this was an exploratory proof-of-concept, adaptive dose optimisation pharmacometric, and safety study in healthy G6PD-deficient males, there was no formal sample size calculation. We reasoned that if the tested primaquine regimens were well tolerated in 20 volunteers (i.e. four cohorts), this would provide preliminary evidence for the safety and the feasibility of this approach. In addition, the rich longitudinal data could then be used to develop an intra-host model to design optimal ascending regimens (*Watson et al., 2017*).

Graphical visualisation of the data was the primary approach to qualitatively characterising the safety of the trialled regimens. For the primary analysis we pre-specified primary and secondary outcomes which are detailed in Appendix 3.

## Statistical analysis

All data analysis was done in R version 4.2.2. The baseline value for each continuous measurement was defined as the mean of the screening and day 0 measurement. Haemoglobin was measured using HemoCue (daily, two samples) and using a laboratory-processed CBC (every 4–5 days). The daily mean haemoglobin concentration was calculated as the mean of the HemoCue derived haemoglobin (the mean of the two samples) and the haemoglobin concentration from the CBC (if no CBC was done then just the mean HemoCue value). The baseline haemoglobin was then calculated as the mean of the values at screening and on day 0. The dose-response models were linear models fit using maximum likelihood. Linear regression was performed to estimate the relationship between the day 10 cumulative primaquine dose and the primary and secondary outcomes (listed in Appendix 3).

# Results

## Study population

Of 215 potential subjects (either identified through hospital records or screened at the walk-in clinic for G6PD deficiency), 27 male hemizygote G6PD-deficient volunteers were enrolled to the two studies between November 2018 and August 2022. In subjects who were interested in participating in the study, there were two screening failures (one unidentified *G6PD* genotype and one elevated AST/ALT), see CONSORT diagrams in Appendix 4.

**Table 1.** Baseline characteristics of the healthy male volunteers.
For the continuous variables we show the median (range). Of the 27 volunteers, 13 participated in both sub-studies.

| | Part 1 - Ascending dose | Part 2 - Single 45 mg dose | Overall |
|---|---|---|---|
| n | 24 | 16 | 27 |
| Age (years) | 32 (18–55) | 34 (20–58) | 32 (18–58) |
| Weight (kg) | 64 (46–86) | 64 (52–86) | 64 (46–86) |
| *G6PD* genotype | | | |
| Viangchan (871G>A) | 12 | 6 | 12 |
| Mahidol (487G>A) | 4 | 2 | 4 |
| Canton (1376G>T) | 4 | 3 | 5 |
| Aures (143T>C) | 1 | 1 | 1 |
| Chinese-4 (392G>T)[†] | 1 | 0 | 1 |
| Orissa (131C>G) | 1 | 1 | 1 |
| Union (1360C>T) | 1 | 2 | 2 |
| Kaiping (1388G>A) | 0 | 1 | 1 |
| G6PD enzyme activity (U/g Hb) | 0.15 (0–1.9) | | |
| Haemoglobin (g/dL) | 14.3 (11.8–15.8) | 14.0 (12.3–15.9) | |
| Red cell count (×10$^{12}$ per L) | 4.9 (4.2–6.0) | 5.1 (3.9–5.9) | |
| Reticulocyte count (%) | 2.4 (1.1–4.0) | 2.4 (1.0–2.9) | |
| Platelet count (×1000 per µL) | 285 (190–424) | 289 (174–412) | |
| Total white blood cell count (×1000 per µL) | 6.6 (4.8–9.3) | 6.6 (5.2–8.4) | |
| Methaemoglobin (%) | 0.5 (0–1.5) | 0.7 (0–1.4) | |
| AST (IU/L) | 23 (15–60) | 21 (14–36) | |
| ALT (IU/L) | 26 (10–85) | 22 (11–47) | |
| Creatinine (mg/dL) | 0.9 (0.8–1.1) | 1.0 (0.7–1.1) | |
| Total bilirubin (mg/dL) | 0.6 (0.3–1.6) | 0.7 (0.3–1.3) | |
| Haptoglobin (g/L) | 1.1 (0.5–1.7) | 1.1 (0.5–1.7) | |
| *CYP2D6* genotypes | | | |
| *10/*10 | 6 | 4 | 8 |
| *2/*10 | 7 | 4 | 7 |
| *1/*10 | 6 | 4 | 7 |
| *1/*2 | 3 | 3 | 3 |
| *1/*1 | 2 | 1 | 2 |
| Haemoglobin E trait[‡] | 7 | 4 | 7 |
| Presumptive alpha-thalassaemia trait* | 5 | 5 | 7 |

*See definition in Methods.
[†]Also known as Quing Yan (***Minucci et al., 2012***).
[‡]Haemoglobin typing done by electrophoresis in Part 1 only.

The COVID-19 pandemic interrupted the end of Part 1 and delayed finishing the study by 2 years, resulting in a substantially longer interval between test regimens than planned. After the lockdowns, fewer than anticipated Part 1 participants could be recontacted for Part 2, so three additional volunteers were recruited. The volunteer baseline summary characteristics are shown in *Table 1*. All volunteers had low to unmeasurable blood G6PD enzyme activity. The most common *G6PD* variant was Viangchan (871G>A, n=12), as it is in much of the eastern Greater Mekong subregion (*Bancone et al., 2019*). This was followed by Canton (1376G>T, n=5) and Mahidol (487G>A, n=4). A large proportion of subjects had screening reticulocyte counts over 2.5%, i.e. above the normal range (8/24 in Part 1 and 3/16 in Part 2). The intermediate *CYP2D6* metaboliser genotype (homozygous *10) with an activity score of 0.5 was identified in 8/27 (30%) of the volunteers (*Caudle et al., 2020*).

## Ascending dose primaquine (Part 1)

Of 24 volunteers assigned ascending dose primaquine regimens, 23 were included in the analysis (*Appendix 4—figure 1*). Volunteer number 18 was withdrawn from the study after 3 days of receiving 10 mg of primaquine daily because of severe low back pain which resulted from an MRI-confirmed prolapsed intervertebral disc that improved with symptomatic treatment. He was not followed up and did not participate in Part 2. As this was considered unrelated to drug administration and the total primaquine dose received was very low, his data were excluded from the primary outcome analysis. There were no changes in his haemoglobin over the 3 days of primaquine dosing. The remaining 23 subjects in the primary analysis population received ascending primaquine dose regimens of between 11 and 20 days' duration.

### Adverse events resulting in study withdrawal

Primaquine was generally well tolerated. There were no serious adverse events or complications. In three subjects (13%), the ascending dose primaquine regimen was stopped; one for excessive haemolysis and two because of abnormal liver function tests (elevated transaminases).

After receiving 11 doses of a 16-day regimen, subject 11 (*G6PD* Union) reached a fractional haemoglobin fall from baseline of 39.5% (8.9 g/dL vs. 14.7 g/dL at baseline), associated with marked fatigue. This decrease met the stopping rule for study withdrawal (dose limiting toxicity, see Appendix 2). He developed a substantial compensatory reticulocytosis (15%). Over the next 5 days, his haemoglobin remained at ~9 g/dL (nadir observed haemoglobin was 8.8 g/dL corresponding to a 40% decrease from baseline) and rose thereafter to 13.0 g/dL (day 28) and 14.9 g/dL by day 49 (*Appendix 5—figure 1*). The haemoglobinuria Hillmen score peaked at 4 on day 10 and was 3 on day 12, dropping back to 1 on day 13 (*Appendix 5—figure 2*).

Volunteer 7, who was receiving a 20-day regimen, developed asymptomatic rises in ALT of 207 U/L (>5 times upper limit of normal [ULN], grade 3) and AST of 89 U/L (>2 times ULN, grade 1) on day 16, associated with a raised LDH (512 IU/L) and a normal direct (0.3 mg/dL) and indirect (0.5 mg/dL) bilirubin. He was hepatitis B immune, hepatitis A IgM negative, but hepatitis E IgM positive with a borderline IgG result, consistent with early hepatitis E infection. A liver ultrasound showed slightly increased liver echogenicity consistent with a mild fatty liver and/or mild hepatic parenchymal disease. His primaquine dosing was stopped on day 15. His increases in AST and ALT were considered most likely a result of asymptomatic hepatitis E and not primaquine. His haemoglobin was 11 g/dL at the time of withdrawal (baseline: 14.3 g/dL; 23% fall); he was subsequently lost to follow up.

Volunteer 14 (16-day regimen) also developed asymptomatic rises in ALT to reach plasma concentrations of 423 U/L (>10 ULN, grade 3) and AST of 229 U/L (>5 times ULN, grade 3) on day 11, associated with a raised LDH (608 IU/L) mildly raised direct (0.4 mg/dL) but normal indirect (0.9 mg/dL) bilirubin. He was also hepatitis B immune, and was hepatitis A and E antibody negative, but had a mild fatty liver on ultrasound. Primaquine dosing was stopped on day 11. By day 28, the ALT was 87 IU/L and the AST was 36 IU/L. As no other cause could be found, his increases in AST and ALT were considered to be probably related to primaquine. His day 11 haemoglobin was also 11 g/dL (baseline: 13.9; 21% fall).

### Adverse events resulting in dose adjustments

In a further three subjects, the intended ascending dose regimen was not completed as they had haemoglobin falls of 30% and 40% relative to baseline. Subject 13 (16-day regimen assigned) did not

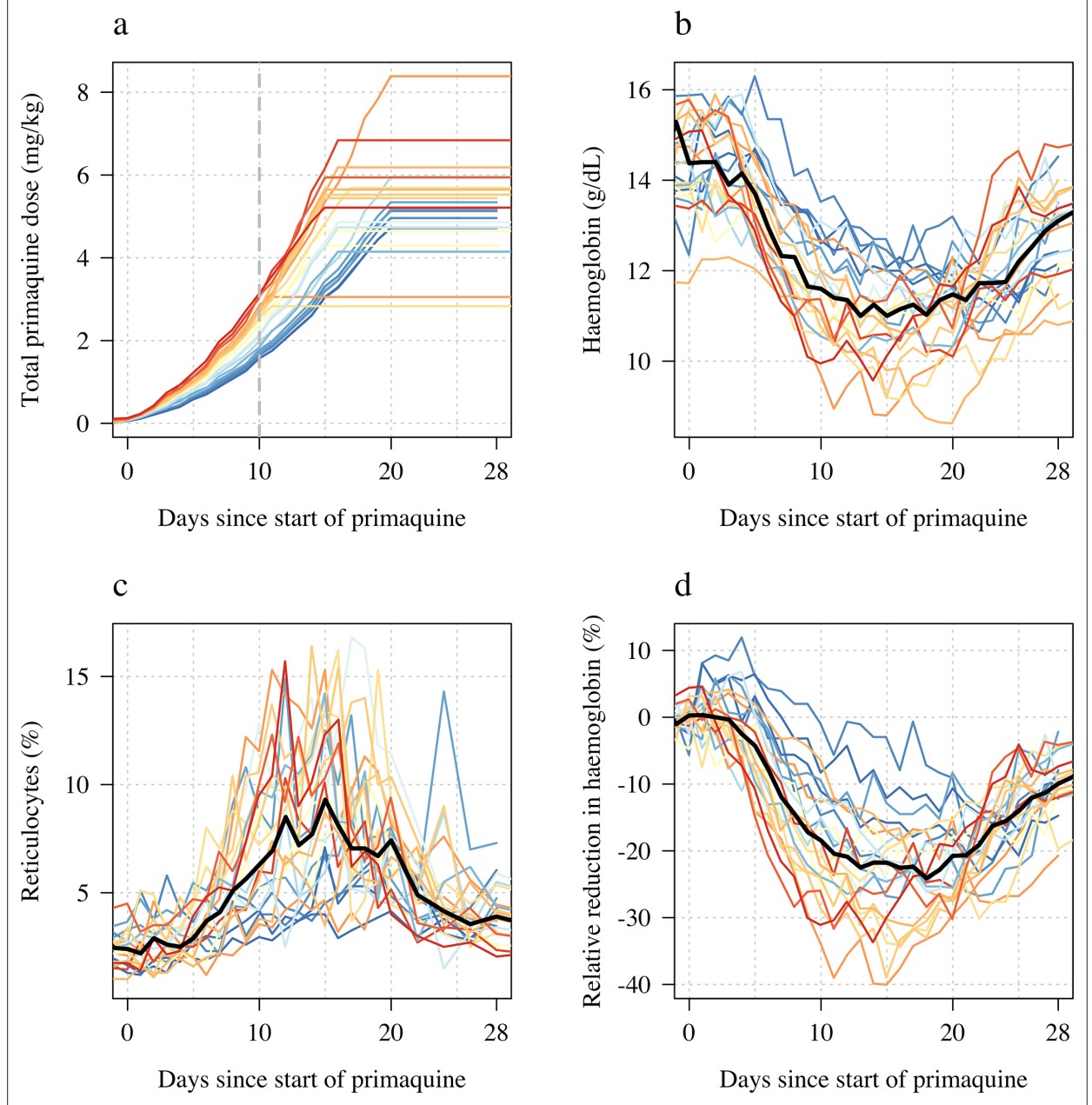

**Figure 1.** Ascending dose study in 23 male hemizygote glucose-6-phosphate dehydrogenase (G6PD)-deficient healthy volunteers included in the primary analysis. Colours from blue to red are in order of increasing day 10 total cumulative primaquine dose as shown in panel **a** (cumulative primaquine doses over time). Panels **b** and **c** show the absolute haemoglobin values and reticulocyte counts over time; panel **d** shows the relative change from baseline in haemoglobin over time. The thick black lines in panels **b–d** show the daily median values.

escalate from 30 to 45 mg on day 15 but stayed at 30 mg until day 16 (33% and 34% fall from baseline haemoglobin on days 15 and 16, respectively). Subject 23 (15-day regimen) was given an additional day at 30 mg on day 12 (day 11 haemoglobin fell 32% from baseline) and escalated to 45 mg on day 14. Subject 24 (15-day regimen) remained at 22.5 mg per day from day 7 to day 15 (instead of escalating to 30 and then 45 mg per day) as his haemoglobin reduction from baseline stayed between 30% and 33%.

## Haemolysis and reticulocyte response

The median absolute fall in haemoglobin from baseline was 3.7 g/dL (range: 2.1–5.9), corresponding to a median relative decrease of 26% (range: 15–40), *Figure 1* panels b and d. The median day of haemoglobin nadir was 16 days after starting primaquine (range: 11–20). There was substantial variation between individuals, including between those with the same *G6PD* genotype who received the same regimens. For example, volunteer 15 (*G6PD* Viangchan) received 6.8 mg/kg total dose of primaquine over 15 days and his haemoglobin dropped around 25% (baseline was 13.6 g/dL; nadir of 10.1 g/dL was reached approximately by day 11) whereas volunteer 20 (also *G6PD* Viangchan) received 5.4 mg/kg over 15 days (slightly faster escalation using the same doses but he was 12 kg heavier), and his haemoglobin fell around 40% (baseline was 15.0 g/dL; nadir of 9.2 g/dL was reached by day 15), see Appendix 6. None of the subjects had a fall of haemoglobin below 8 g/dL and none developed frank haemoglobinuria (Hillmen score ≥6). Peak reticulocytosis occurred at approximately

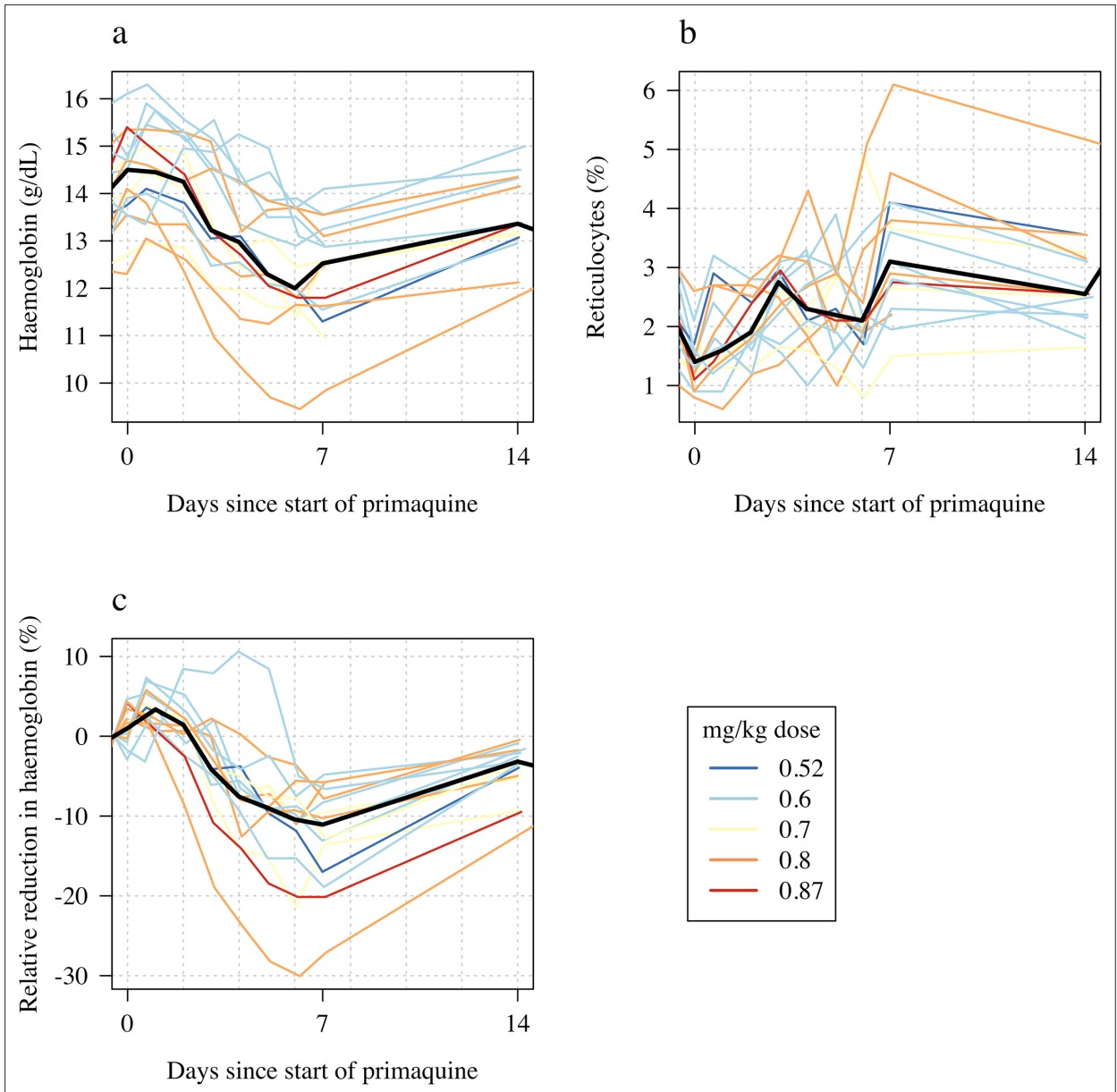

**Figure 2.** Haemolytic effect of 45 mg single primaquine dose in 16 male hemizygote glucose-6-phosphate dehydrogenase (G6PD)-deficient healthy volunteers. Colours from blue to red are in order of increasing mg/kg dose. Panel **a**: absolute haemoglobin values; panel **b**: reticulocyte counts; panel **c**: relative change in haemoglobin from baseline. The thick black lines in panels a–c show the daily median values.

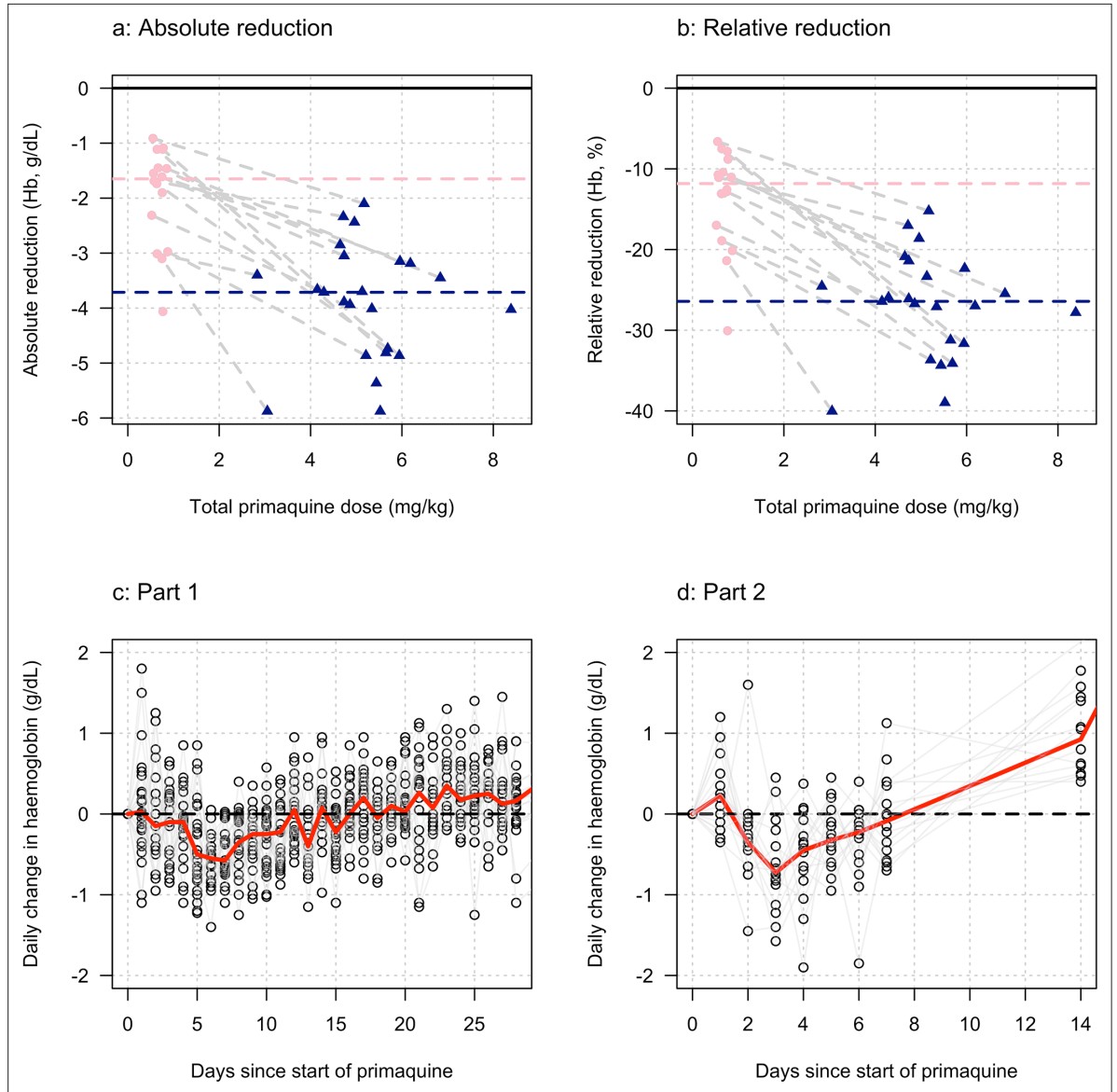

**Figure 3.** Comparing the haemolytic effect of ascending dose primaquine regimens (dark blue triangles) and the single 45 mg primaquine dose (pink circles). Panels a-b show the relationship between the total cumulative dose of primaquine given to each subject in each study (x-axis) and the absolute fall in haemoglobin concentration (panel a) or the relative fall (panel b). Subjects who participated in both parts are joined by the light grey dashed lines. The horizontal dashed lines show the median falls by sub-study. Panels c-d show the daily observed changes in haemoglobin (c: Part 1; d: Part 2), the red line shows the daily median change.

the same time as the haemoglobin nadir (day 16; range: 11–20), with a median peak reticulocyte count of 10.3% (range: 4.2–16.8), *Figure 1* panel c. It took approximately 2 weeks for the reticulocyte counts to re-normalise.

## Single 45 mg primaquine dose (Part 2)

Following a single 45 mg base equivalent dose of primaquine (mg base/kg doses ranged from 0.52 to 0.87), there was a marked fall in haemoglobin concentrations reaching a median observed nadir on day 6 (range: days 4–7). The median fall from baseline was 1.7 g/dL (range: 0.9–4.1 g/dL decrease), corresponding to a median relative decrease of 12% (range: 7–30% decrease), see *Figure 2*. The largest daily falls in haemoglobin following the single dose were on day 3 (median fall in haemoglobin on day 3 was 0.73 g/dL, *Figure 3*).

The reticulocyte response was characterised by a gradual rise with most volunteers having their observed peak reticulocyte proportions on day 7 (range days 4–14). The largest fall in haemoglobin, and greatest rise in reticulocyte count occurred in subject 13 who was *G6PD* Canton (absolute fall of 4.1 g/dL: baseline haemoglobin was 13.6 g/dL, the nadir haemoglobin was 9.5 g/dL on day 7; relative fall of 30%). He did not participate in Part 1 so there are no ascending dose data with which to compare.

## Comparison of ascending and single dose primaquine regimens

*Figure 3* shows the maximum observed absolute and relative falls in haemoglobin concentration as a function of the total primaquine dose received across the two studies. Overall, the median fall in the single dose cohort was nearly half that of the median fall in the ascending dose group, whereas the median total dose in the single dose cohort was only 14% of the median total dose in the ascending dose cohort. Although the subjects receiving a single 45 mg dose of primaquine had smaller absolute falls in haemoglobin, they experienced earlier and greater daily falls (*Figure 3* panels c-d). The greatest median falls in haemoglobin in Part 1 were on days 6 and 7 (0.55 and 0.57 g/dL reductions, respectively), whereas the median fall on day 3 (day with the largest median fall) in the single dose cohort was 0.73 g/dL. Two subjects receiving the 45 mg single dose had daily haemoglobin falls of nearly 2 g/dL.

## Haemolysis dose-response relationship

We summarised each ascending dose regimen in Part 1 by the cumulative dose of primaquine received by day 10. This summary exposure statistic was chosen following graphical visualisation of the haemoglobin data, which showed that the most substantial haemolysis had occurred by day 10 of the study (*Figure 1b*, note that the nadir haemoglobin was observed after day 10 in all volunteers). In the 23 volunteers, the day 10 cumulative dose varied from 1.7 to 3.5 mg/kg, with a median value of 2.6 mg/kg.

*Figure 4* shows the dose-response data coloured by *G6PD* variant. The day 10 cumulative dose was predictive of both the maximum absolute fall in haemoglobin with respect to the baseline value (1.2 g/dL fall per mg/kg increase [95% CI: 0.5–1.8]; p=0.001, $r^2 = 0.40$), and the maximum relative fall in haemoglobin with respect to the baseline value (7.9% fall per mg/kg increase [95% CI: 4.4–11.3]; p=0.0002; $r^2 = 0.49$), but not of the average daily fall over days 5–10 (0.06 g/dL per day [95% CI: 0.14 to –0.03]), *Figure 4* panels a, c and e. For Part 2 the mg/kg dose was not associated significantly with either the absolute, relative, or the mean daily falls in haemoglobin, although for all three outcomes the point estimates were in the expected direction (i.e. greater mg/kg doses resulting in larger falls), *Figure 4* panels b, d and f.

In an additional exploratory analysis, there was no evidence of substantial differences in haemolysis between the different *G6PD* variants (although the sample size is very small). There was some evidence that subjects with higher baseline haemoglobin concentrations had larger relative falls. There was no evidence for an association between the homozygous *10 *CYP2D6* genotype associated with reduced primaquine bioactivation and haemolysis (i.e. no evidence that these subjects haemolysed less than the other subjects).

## Effect on total plasma bilirubin and LDH

The falls in haemoglobin were associated with predictable biochemical changes indicative of haemolysis. There were rises in plasma concentrations of the intraerythrocytic enzyme LDH and in total bilirubin reflecting haem metabolism. The more rapid fall in haemoglobin associated with the single 45 mg primaquine dose was associated with larger normalised rises in total bilirubin consistent with greater haemolysis (*Figure 5* and Appendix 7).

## Liver transaminases, creatinine, haptoglobin, and methaemoglobin

In both studies, there was no evidence for a relationship between primaquine dose (Part 1: cumulative day 10 total mg/kg dose; Part 2: single mg/kg dose) and maximum observed fold change in serum AST, ALT, or plasma creatinine, or the maximum observed absolute decrease in plasma haptoglobin (Appendix 8). There were no clinically significant rises in blood methaemoglobin in any of the study participants (*Figure 6* panels a–b). In Part 1, the day 10 cumulative dose was negatively associated

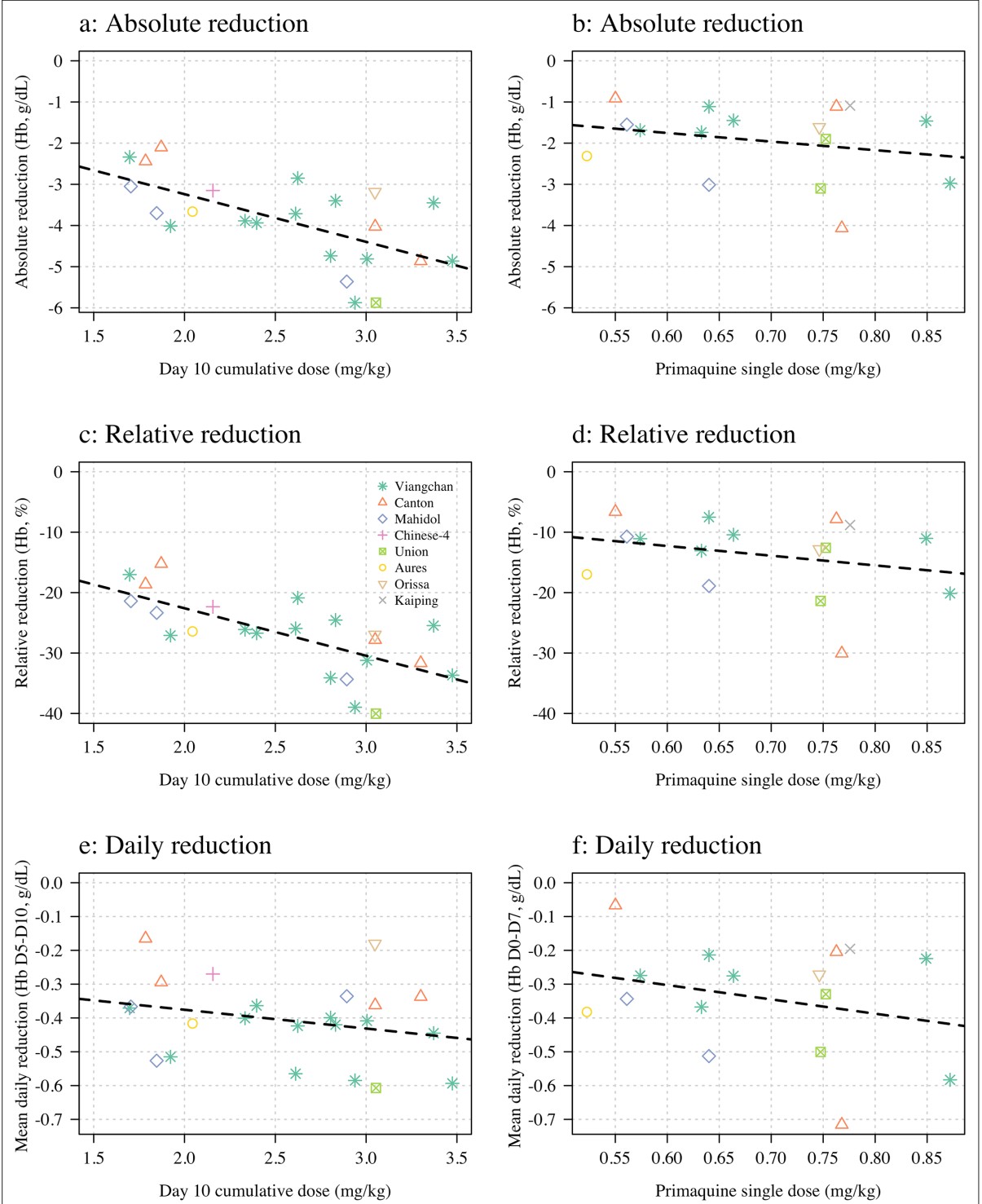

**Figure 4.** Haemolysis dose-response relationships (panels **a, c** and **e** correspond to Part 1; panels **b, d**, and **f** to Part 2). Three pre-specified outcome variables are shown: the maximum absolute reduction in haemoglobin (panels **a & b**); the maximum relative reduction (panels **c & d**); and the mean daily reduction between days 5 and 10 (panels **e & f**). The dose exposure summary in Part 1 is the day 10 cumulative primaquine dose (n=23); for Part 2 it is the mg/kg single dose (n=16). Hb: haemoglobin.

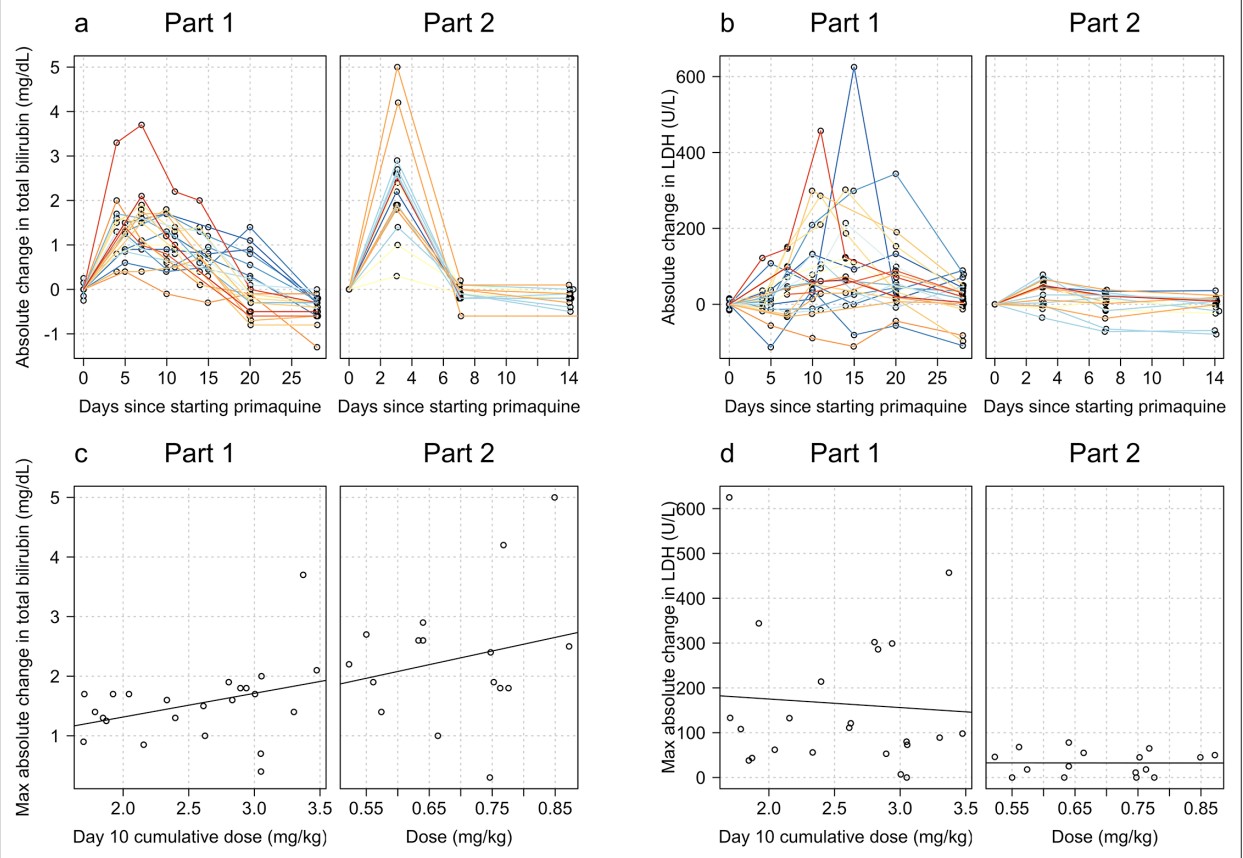

**Figure 5.** Relationship between primaquine dose (Part 1: day 10 cumulative mg/kg dose; Part 2: mg/kg single dose) and total plasma bilirubin (panels **a** and **c**) and lactate dehydrogenase (LDH) concentrations (panels **b and d**). The top rows show the normalised data; the bottom rows the dose-response relationship with the maximum normalised increases.

with the peak observed methaemoglobin concentration (*Figure 6* panel c, p=0.02) with a similar negative trend was also observed in Part 2 (*Figure 6* panel d; p=0.06). There was no association between having a poor metaboliser *CYP2D6* genotype (*10 homozygous versus other genotypes) and peak blood methaemoglobin concentration.

## Discussion

The 8-aminoquinoline antimalarials are the only drugs which prevent relapses of vivax or ovale malaria (radical cure). Significant haemolysis from radical cure regimens cannot be avoided in G6PD deficiency but, using the rapidly eliminated primaquine, it can be attenuated. Ascending dose primaquine regimens in G6PD-deficient malaria patients exploit the same pharmacodynamic principle underling the current once weekly treatment recommendation (*Alving et al., 1960*). They aim to provide controlled haemolysis while allowing for steady reconstitution of the red cell population with increasingly younger, and therefore 'oxidant-resistant' erythrocytes (*Kellermeyer et al., 1962*). As G6PD testing is usually unavailable in malaria endemic areas, the prescriber treating vivax or ovale malaria is currently faced with the therapeutic dilemma of either not giving the drug, and failing to prevent relapses with their attendant substantial morbidity, or giving it and causing iatrogenic haemolysis in G6PD-deficient patients. The net result is that radical cure primaquine regimens are often not prescribed, even though they would be well tolerated and safe in the majority of patients who are G6PD-normal. Primaquine underuse is a major contributor to global vivax malaria morbidity.

The currently recommended 8-week radical cure regimen for G6PD-deficient patients attenuates the fall in haemoglobin, but it still risks significant haemolysis, particularly with the first 0.75 mg/kg dose (*Kheng et al., 2015*). This regimen has not been well evaluated in patients with severe *G6PD*

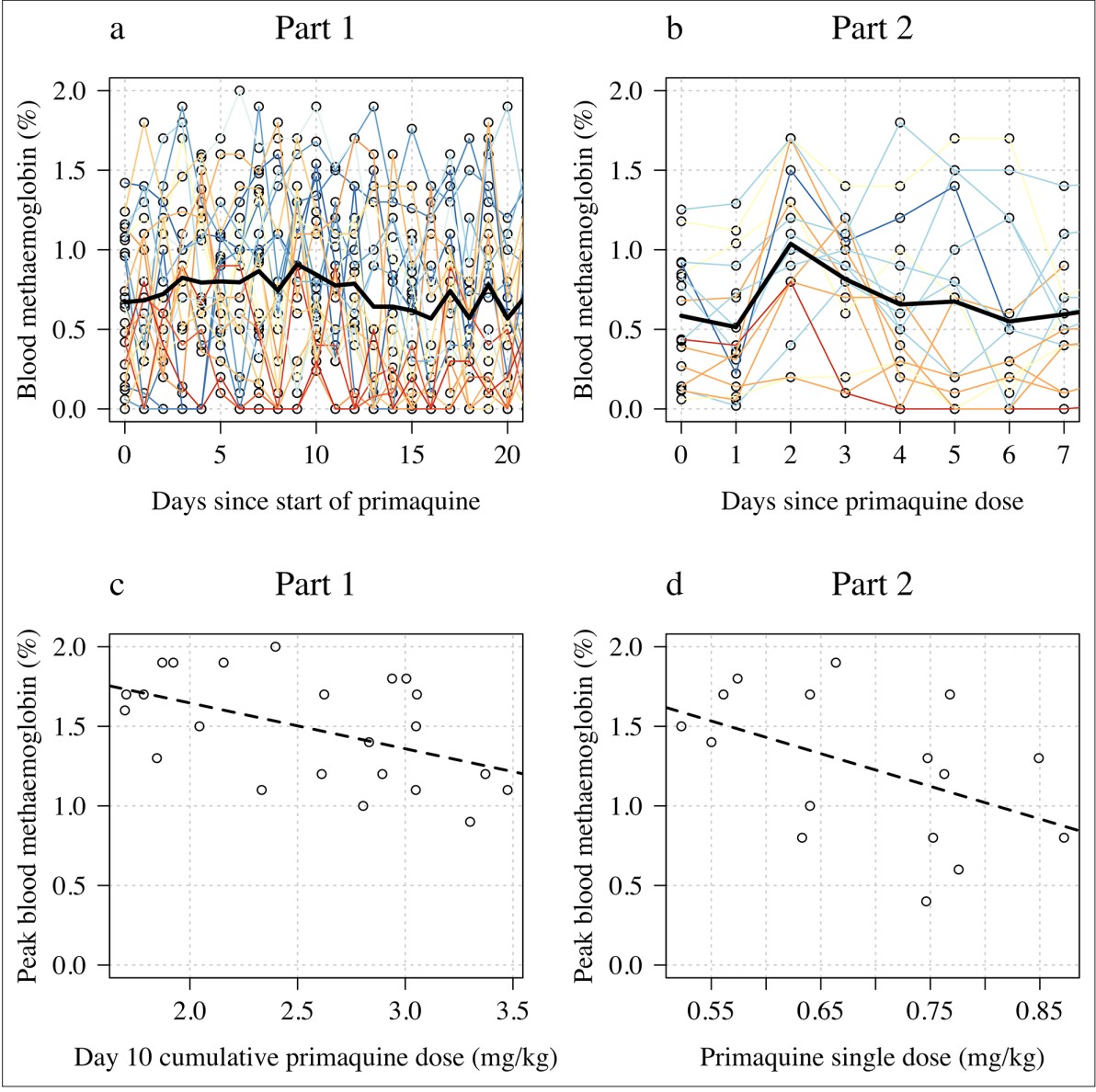

**Figure 6.** Changes in blood methaemoglobin concentration. Top row: individual data (panel **a**: Part 1; panel **b**: Part 2) with the daily mean value shown by the thick black line. Bottom row: relationship between dose (panel **c**: Part 1 summarised by the day 10 total dose; panel **d**: Part 2 summarised by the mg/kg dose) and peak observed blood methaemoglobin (%).

deficiency variants who are at greatest risk, and it requires good adherence for 8 weeks. As much of the oxidant haemolysis occurs after the first dose, failure to complete the 8-week course (which is likely to be common) therefore incurs most of the haemolytic risk without providing the full benefit.

In this study the single 45 mg dose resulted in a median fall in haemoglobin concentration of 1.7 g/dL, which was nearly half the median fall observed with the full ascending dose regimen. In comparison, the ascending dose regimens gave a seven times higher total dose. These ascending dose primaquine regimens were relatively well tolerated in adult male volunteers with Southeast Asian variants of G6PD deficiency. Although these *G6PD* variants are generally regarded as moderate in severity, there is wide variation in the phenotype between variants and also between individuals with the same genotype. The ascending primaquine dose regimens are still associated with a clinically significant haemolytic risk. Thus, in considering the radical cure treatment of vivax malaria, the risks of relapse, often on multiple occasions with consequent anaemia, must be balanced against the predictable haemolysis that will result from the oxidative effects of the 8-aminoquinoline radical cure regimen (*Commons et al., 2020*; *Commons et al., 2019*). In the Southeast Asian region approximately half the vivax

malaria cases will be followed by at least one relapse if radical curative treatment is not given. Many factors need to be incorporated in the assessment of both population and individual risk from haemolysis. Acute kidney injury only results from fulminant haemolysis. The main risk is dangerous anaemia. At a population level the overall risks are lower than predicted from gene frequencies because severe variants of G6PD deficiency protect against vivax malaria, although whether they affect the risk of relapse is not known (*Awab et al., 2021*). At an individual level the haemolytic risk depends on the *G6PD* genotype, the degree of exposure to the putative oxidative 8-aminoquinoline metabolites, and also the degree of pre-existing anaemia. Vivax malaria causes anaemia, but as the haemolytic component of the malarial anaemia also results in loss of older, more deficient erythrocytes, the fractional reduction in erythrocytes resulting from the oxidant drug in acute malaria is correspondingly less. Finally, chronic anaemias associated with haemolysis (e.g. hookworm, recurrent malaria) result in younger red cell populations with right shifted oxygen dissociation curves and thus increased oxygen delivery. As a result, the pathological consequences of causing severe anaemia (haemoglobin concentration <5 g/dL) depend on whether the patient was already significantly anaemic before the malaria infection and why they were anaemic.

In this exploratory study, in which there were cautious stopping rules, 3 out of 23 (13%) of the volunteers did not complete the regimen because of drug toxicity; one had significant haemolysis and two had asymptomatic hepatitis. Hepatitis is a rare adverse effect of primaquine (*Recht et al., 2014*). Asymptomatic hepatitis E is well documented and more common than symptomatic disease (*Kamar et al., 2012*). Three volunteers were not escalated to higher primaquine doses as intended because of significant falls in haemoglobin. Such falls in haemoglobin were expected by design in this exploratory adaptive study. In none of the volunteers did haemoglobin concentrations fall below 8 g/dL. The extent of haemolysis varied substantially between volunteers, even within the same genotype. Higher baseline haemoglobin values tended to be followed by greater falls. Taken together with the high baseline reticulocytosis (*Bancone et al., 2017*), this suggests that subjects with G6PD deficiency have variably shortened red cell survival, and thus variably sized populations of erythrocytes vulnerable to oxidant haemolysis.

One of the important advantages of primaquine (half-life approximately 5–7 hr) relative to its slowly eliminated analogue tafenoquine (half-life approximately 15 days) is that the treatment can be stopped as soon as there are signs or symptoms of haemolytic toxicity. Despite the use of primaquine for nearly 70 years and administration of hundreds of millions of treatments there have been very few reported deaths from haemolytic toxicity (*Recht et al., 2014*; *Yilma et al., 2023*). Very large mass treatments with primaquine have been used in malaria elimination campaigns involving millions of people in China, Nicaragua, Turkmenistan, Azerbaijan, Tajikistan, Afghanistan, and North Korea (*Hsiang et al., 2013*; *Kondrashin et al., 2014*). In the latter four countries an interrupted regimen involved giving 4 days of primaquine, stopping for 3 days and then completing a further 10 days. Subjects with significant haemolysis could stop the drug at any time. Serious toxicity was very rare, despite the high prevalences of G6PD deficiency in some of the regions. In West and Central Asia the severe *G6PD* Mediterranean variant would have been the most prevalent genotype.

In G6PD-deficient patients with acute malaria, the disease itself causes haemolytic anaemia (*White, 2018*), and the consequent preferential loss of older erythrocytes ameliorates the adverse impact of oxidant drugs. On the other hand, compensatory reticulocytosis might be inhibited but limited experience with weekly primaquine shows a robust reticulocyte response post treatment (*Kheng et al., 2015*). Differences in haemolytic response between healthy volunteers and malaria patients are likely to be small as the illness usually resolves within a few days with effective treatment, while the ascending dose regimen would still be using the lowest dose.

The main limitation of this therapeutic approach is its complexity. This could be addressed by preparation of blister packed primaquine allowing easy dose transition. The *G6PD* deficiency genotypes studied here are representative of those present in the Southeast Asian region, and can be regarded as of moderate severity, with the African A-genotype (in which the currently recommended once weekly dosing regimen was developed) being at the less severe end of the spectrum, and the common Mediterranean *G6PD* genotype being at the more severe end. The safety of this ascending dose regimen in patients with severe *G6PD* deficiency cannot be predicted based on these data. From a therapeutic perspective, as G6PD testing is usually not available in malaria endemic areas, the individual patient risk assessment must take into account the factors described earlier (i.e. the prevalence

of G6PD deficiency and its likely severity, the sex of the patient, the degree of anaemia, and the probability of relapse), and also an assessment of the patients' understanding of the risks and when to stop treatment, and the likelihood and feasibility of accessing health care if there is severe haemolysis.

In summary, shorter course ascending dose vivax malaria radical cure regimens in G6PD-deficient subjects offer the prospect of an effective treatment which does not incur prohibitive haemolytic toxicity and in some areas could obviate the need to test for G6PD deficiency.

## Data sharing statement

All analysis code and data are available via an accompanying github repository: https://github.com/jwatowatson/Primaquine-Challenge.

## Acknowledgements

We are very grateful to the volunteers who participated in this study. We thank the staff of the Clinical Therapeutics unit and the laboratories in the Hospital for Tropical Diseases, Bangkok, which provided essential monitoring. We thank the Independent Drug Safety Monitoring Board members: Prof. Asim Beg (chair), Dr. Bushra Moiz, and Prof. Rajitha Wickremasinghe. This study was funded by the MRC 'Assessing the tolerability of a potentially safer radical curative regimen of primaquine in healthy volunteers with glucose-6-phosphate dehydrogenase' (MR/R015252/1, grant held by WRT). This UK-funded award is part of the EDCTP2 programme supported by the European Union. NJW is a Principal Research Fellow funded by the Wellcome Trust (093956/Z/10/C). JAW is a Sir Henry Dale Fellow funded by the Wellcome Trust (223253/Z/21/Z). This research was partly funded by Wellcome. A CC BY or equivalent licence is applied to the author accepted manuscript arising from this submission, in accordance with the grant's open access conditions.

## Additional information

### Funding

| Funder | Grant reference number | Author |
|---|---|---|
| Medical Research Council | MR/R015252/1 | Sasithon Pukrittayakamee |
| Wellcome Trust | 10.35802/093956 | Nicholas J White |
| Wellcome Trust | 10.35802/223253 | James A Watson |

The funders had no role in study design, data collection and interpretation, or the decision to submit the work for publication. For the purpose of Open Access, the authors have applied a CC BY public copyright license to any Author Accepted Manuscript version arising from this submission.

### Author contributions

Sasithon Pukrittayakamee, Resources, Supervision, Funding acquisition, Validation, Project administration, Writing – review and editing; Podjanee Jittamala, Resources, Investigation, Methodology, Project administration, Writing – review and editing; James A Watson, Formal analysis, Funding acquisition, Validation, Visualization, Writing - original draft, Writing – review and editing; Borimas Hanboonkunupakarn, Pawanrat Leungsinsiri, Investigation, Methodology, Writing – review and editing; Kittiyod Poovorawan, Validation, Investigation, Methodology, Writing – review and editing; Kesinee Chotivanich, Resources, Investigation, Writing – review and editing; Germana Bancone, Cindy S Chu, Resources, Validation, Writing – review and editing; Mallika Imwong, Resources, Methodology, Writing – review and editing; Nicholas PJ Day, Resources, Supervision, Funding acquisition, Writing – review and editing; Walter RJ Taylor, Resources, Supervision, Funding acquisition, Investigation, Methodology, Project administration, Writing – review and editing; Nicholas J White, Conceptualization, Supervision, Funding acquisition, Validation, Methodology, Writing - original draft, Writing – review and editing

### Author ORCIDs

James A Watson http://orcid.org/0000-0001-5524-0325

Kittiyod Poovorawan http://orcid.org/0000-0001-7016-7605
Cindy S Chu http://orcid.org/0000-0001-9465-8214
Nicholas PJ Day http://orcid.org/0000-0003-2309-1171
Nicholas J White http://orcid.org/0000-0002-1897-1978

### Ethics

Clinical trial registration TCTR20170830002; TCTR20220317004.

The two parts of this study were approved as separate studies. Both parts were approved by the Faculty of Tropical Medicine's Ethics Committee (MUTM 2017-036-01 and MUTM 2021-031-02) and the Oxford Tropical Research Ethics Committee (OxTREC, number 48-16).

Joint Public Review: https://doi.org/10.7554/eLife.87318.3.sa1
Author Response https://doi.org/10.7554/eLife.87318.3.sa2

---

## Additional files

### Supplementary files

• MDAR checklist

### Data availability

All analysis code and data are available via an accompanying github repository: https://github.com/jwatowatson/Primaquine-Challenge (copy archived at *Watson, 2024*).

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

## Appendix 1

### Exclusion criteria

Exclusion criteria were any one of the following:

1. body mass index ≥35;
2. G6PDMed variant (563C>T);
3. known to have, or discovered by the investigator to have any clinically significant disease;
4. malaria or other febrile illness (e.g. viral hepatitis, typhoid fever) in the previous month;
5. a positive blood film for malaria (asexual or sexual parasites);
6. history of haemolysis unrelated to primaquine in the past 8 weeks;
7. blood group Rhesus negative (rare in Thai population and so these subjects were excluded to avoid delays in finding matched blood if needed for transfusion);
8. received a blood transfusion in the past 3 month;
9. blood donation >300 mL of whole blood previous 3 months;
10. taking or taken within the past 3 weeks any herbal medicine or any drug or foodstuff known to cause haemolysis in G6PD deficiency.
11. AST or ALT or LDH >1.5 times the ULN;
12. serum creatinine >ULN (1.2 mg/dL) and an eGFR <70 mL/min/1.73 m² (Chronic Kidney Disease Epidemiology Collaboration [CKD-EPI] equation; *Levey et al., 2009*);
13. dipstick urine analysis showing microscopic haematuria (≥5 RBCs) microscopic and/or microscopic proteinuria;
14. conjugated or unconjugated bilirubin >1.5 ULN;
15. methaemoglobin level >5% determined by oximetry (Masimo Rad 57);
16. allergy to primaquine;
17. having taken part in research involving an investigational drug within the past 8 weeks;
18. potential non-compliance with the protocol by the volunteer (investigator opinion).

# Appendix 2

## Adaptive rules

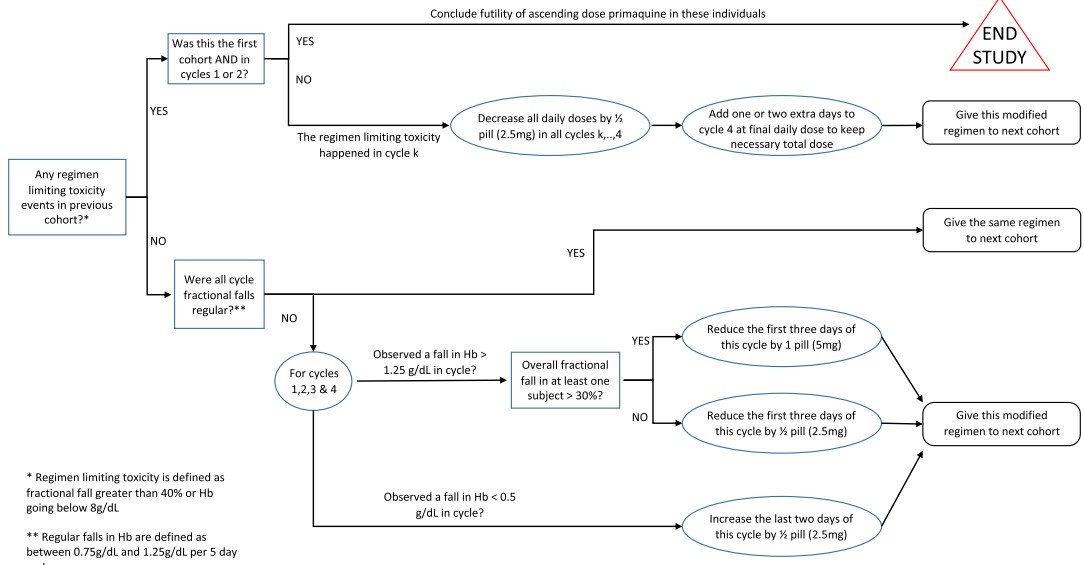

**Appendix 2—figure 1.** Rules for determining the primaquine regimen in the next cohort of five subjects. These pre-specified rules were for guidance only and were not binding as participant safety was the primary concern.

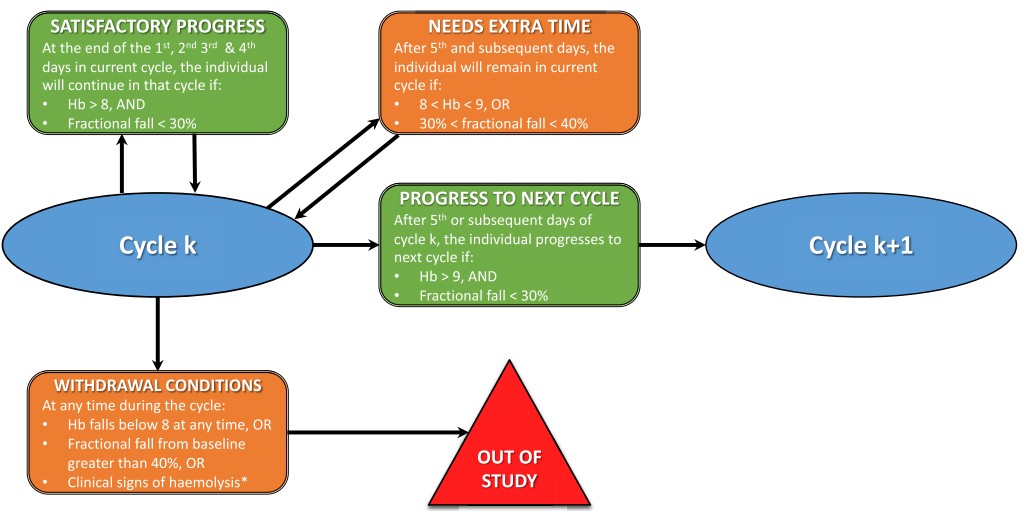

**Appendix 2—figure 2.** Rules for increasing primaquine dose for each subject. Each cycle was between 3 and 5 days, and doses were increased across the successive cycles. Each subject progressed to the next cycle (i.e. higher dose) only if they had a haemoglobin >70% of their baseline value (i.e. <30% relative decrease). Primaquine was stopped if the haemoglobin went under 40% of the baseline value.

## Appendix 3

### Study endpoints

This trial was designed to characterise the dose-response relationship for primaquine-induced haemolysis in healthy adult males with G6PD deficiency. The therapeutic aim was to develop a primaquine radical cure regimen that was safer in G6PD deficiency and shorter than the currently recommended once weekly 0.75 mg base/kg regimen given for 8 weeks. There was no formal hypothesis testing. The statistical analysis plan specified multiple endpoints which summarise the overall safety of the ascending primaquine doses and allow comparison with the single dose study (in particular the haemolytic effects of each).

### Primary outcomes

1. Maximum absolute fall in haemoglobin between day 0 and day 28 (absolute difference with respect to baseline haemoglobin);
2. Maximum relative fall in haemoglobin between day 0 and day 28 (relative change (%) with respect to baseline haemoglobin);
3. Maximum increase in total bilirubin between day 0 and day 28 (absolute difference with respect to baseline total bilirubin);
4. Maximum increase in LDH between day 0 and day 28 (absolute difference with respect to baseline LDH);
5. Mean decrease in haemoglobin per day (g/day) between day 5 and day 10 (Part 1 only: estimated from a linear model fit to daily haemoglobin measurements);
6. Mean decrease in haemoglobin per day (g/day) between day 1 and day 7 (Part 2 only: estimated from a linear model fit to daily haemoglobin measurements).

### Secondary outcomes

1. Maximum fold change relative to baseline for serum AST, ALT, and creatinine;
2. Maximum absolute change in serum haptoglobin;
3. Peak reticulocyte count;
4. Peak blood methaemoglobin percentage;
5. Clinically significant changes in the differential white blood cell count.

## Appendix 4

### CONSORT diagrams

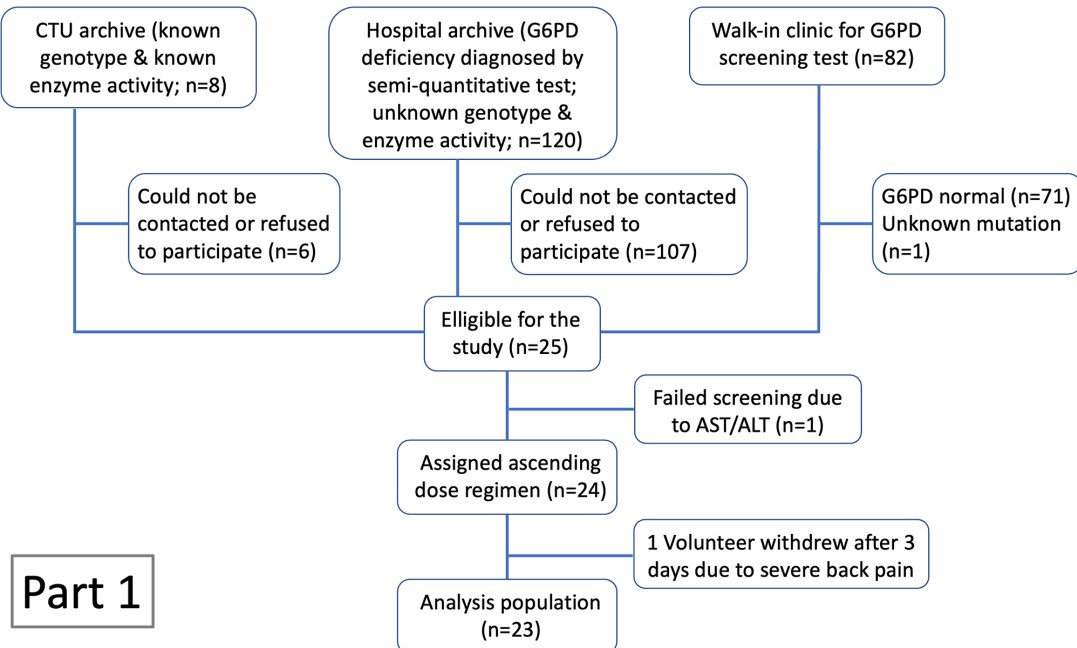

**Appendix 4—figure 1.** CONSORT diagram for Part 1 of the study.

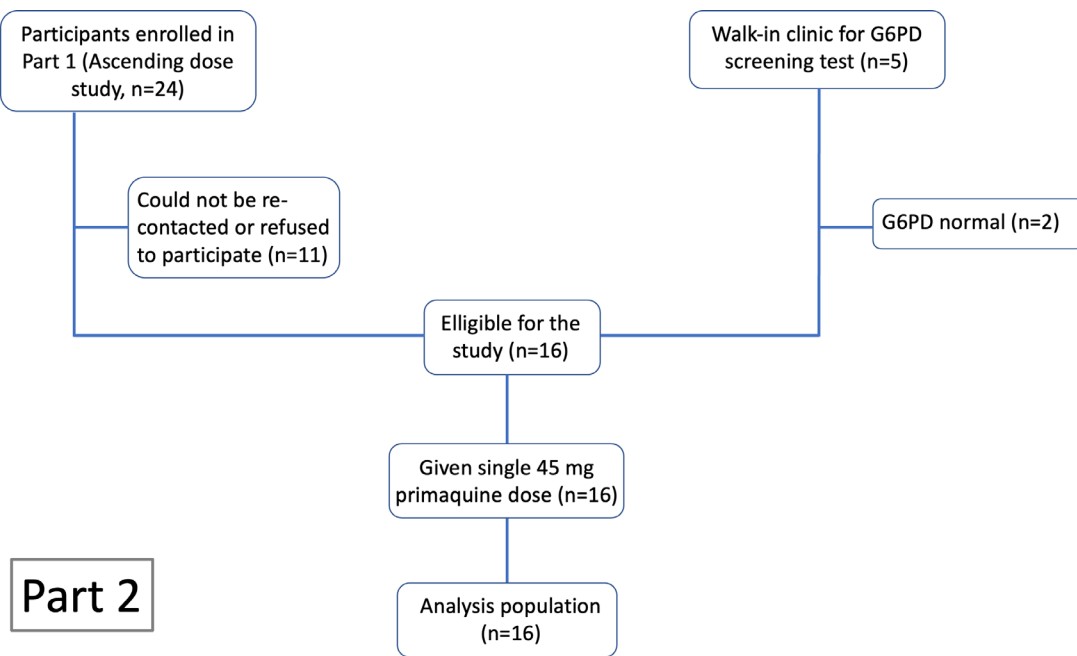

**Appendix 4—figure 2.** CONSORT diagram for Part 2 of the study.

## Appendix 5

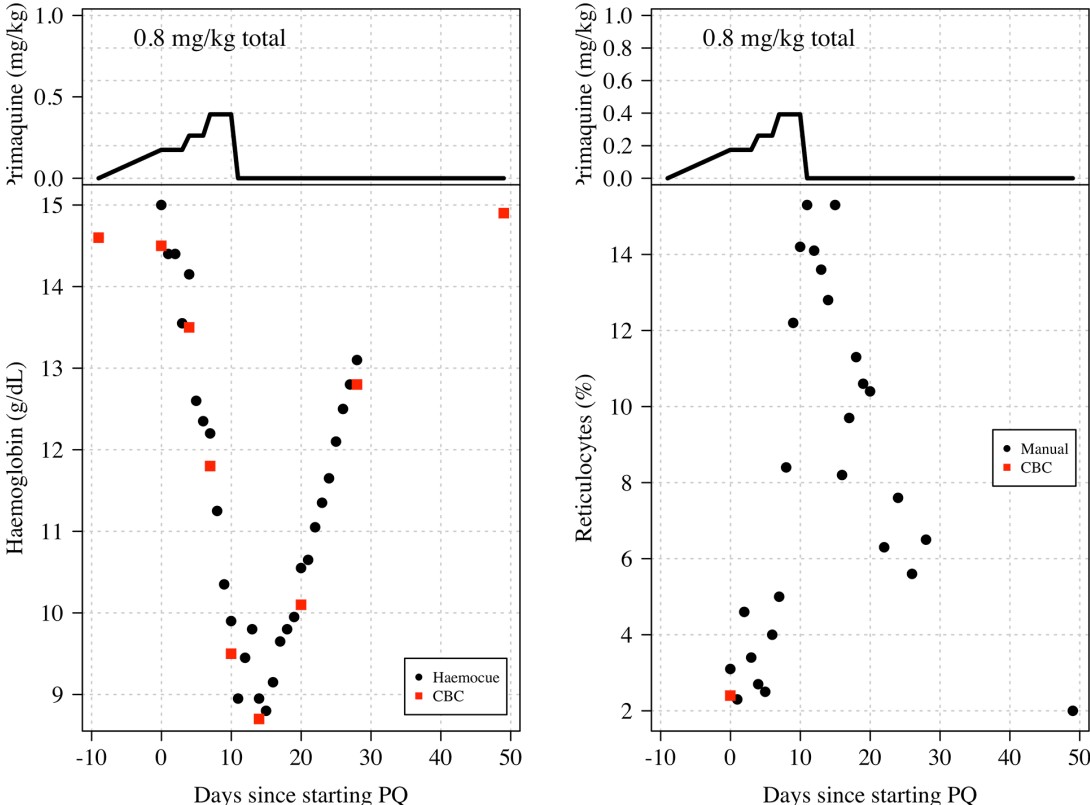

**Appendix 5—figure 1.** Haemoglobin and reticulocyte data from subject 11 who was stopped because he met the study withdrawal stopping rule. The top panels show the dose of primaquine given each day; the bottom panels show the observed haemoglobin measurements (left) and reticulocyte counts (right). Black circles: haemocue and manual reticulocyte counts; red squares: complete blood count (CBC) haemoglobin and reticulocyte counts.

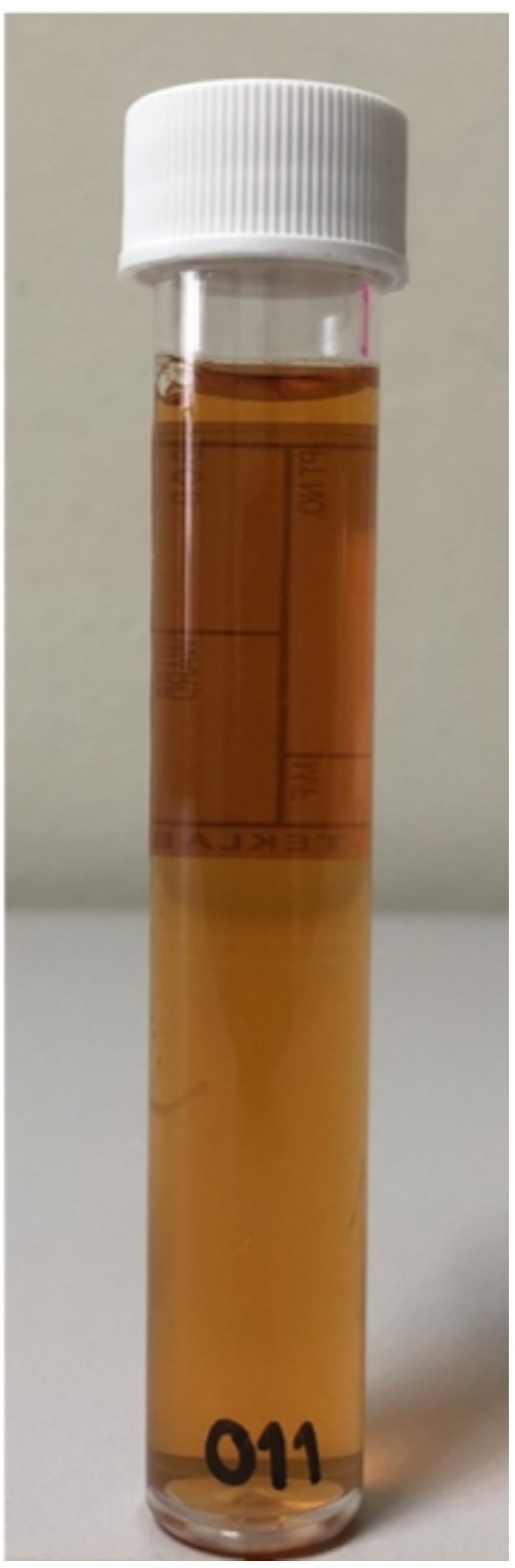

**Appendix 5—figure 2.** Day 10 urine sample from subject 11 showing slight haemoglobinuria (Hillmen score of 4). The subject's maximum Hillmen score on days 4–9 varied between 2 and 3 .

## Appendix 6

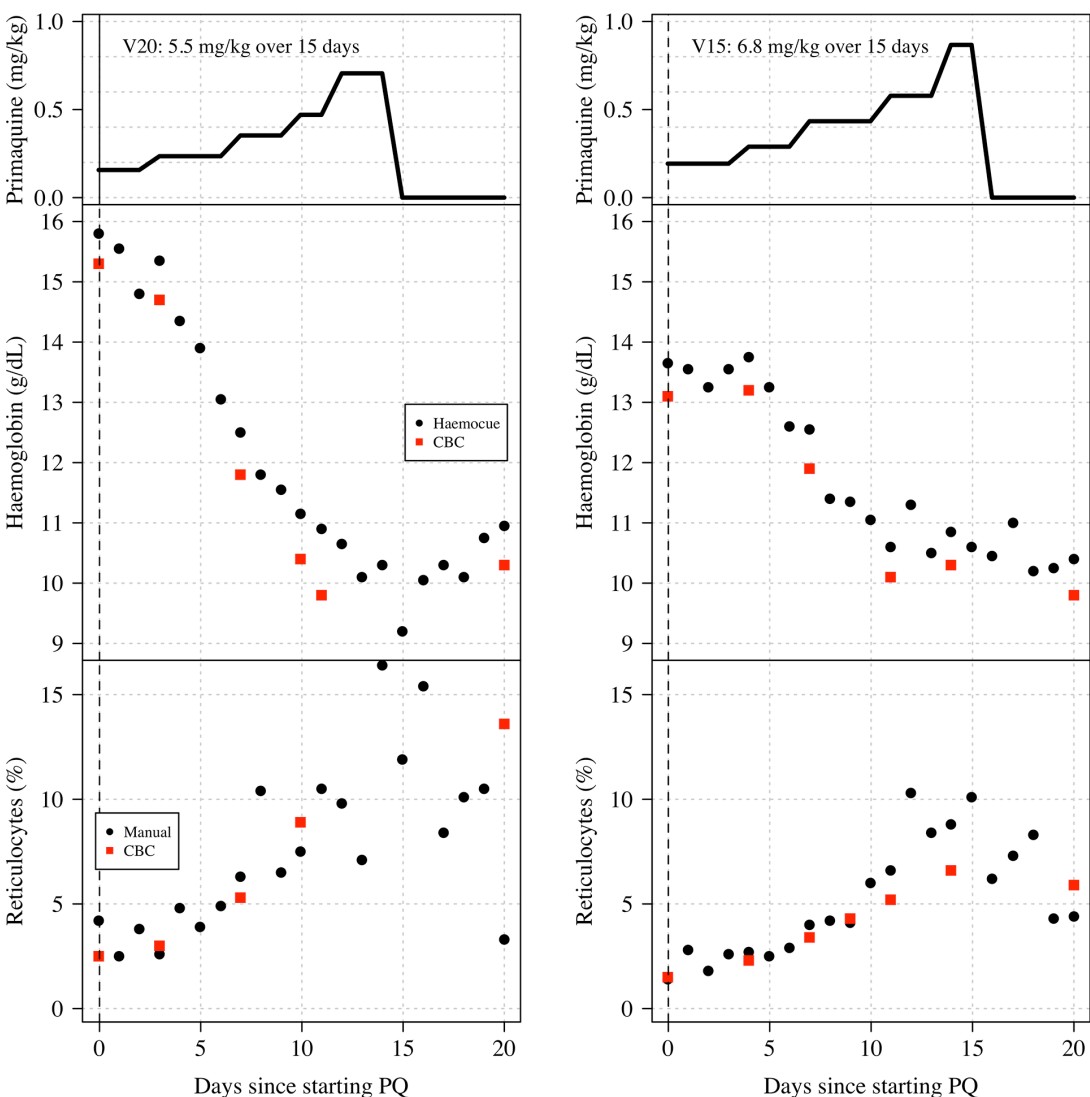

**Appendix 6—figure 1.** Haemoglobin and reticulocyte data from two subjects (left: V15; right: V20) who both had the Viangchan variant (871G>A). Both received very similar ascending dose regimens which are shown in the top panels (mg/kg primaquine daily dose); the middle panels show the observed haemoglobin measurements; the bottom panels show the reticulocyte counts. Black circles: haemocue and manual reticulocyte counts; red squares: complete blood count (CBC) haemoglobin and reticulocyte counts.

## Appendix 7

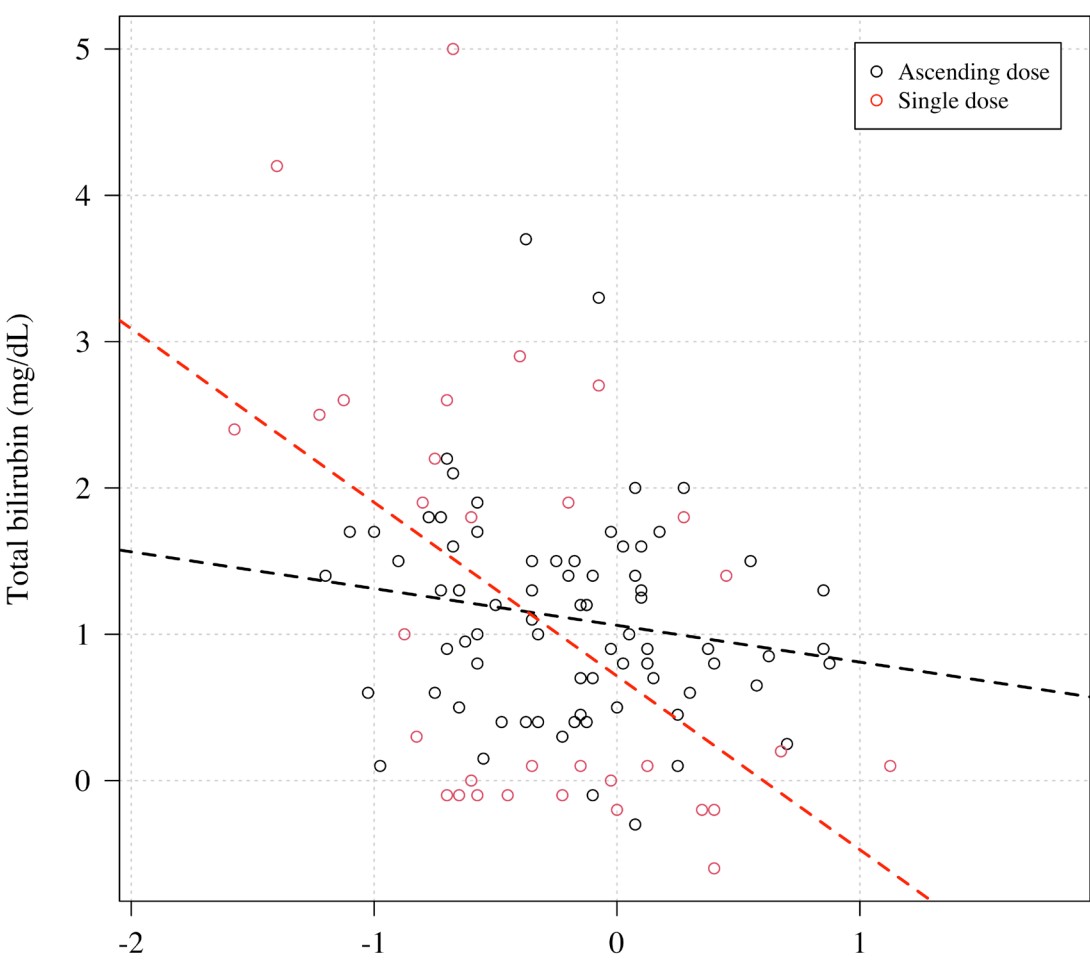

**Appendix 7—figure 1.** The relationship between the reduction in haemoglobin from the previous day and the normalised absolute increase in total bilirubin. Black: Part 1; red: Part 2. Each subject has two datapoints corresponding to when the total bilirubin was measured (days 5 and 10 for Part 1; days 3 and 7 for Part 2). The relationship is significant for Part 2 (p=0.003), not significant for Part 1 (p=0.2).

## Appendix 8

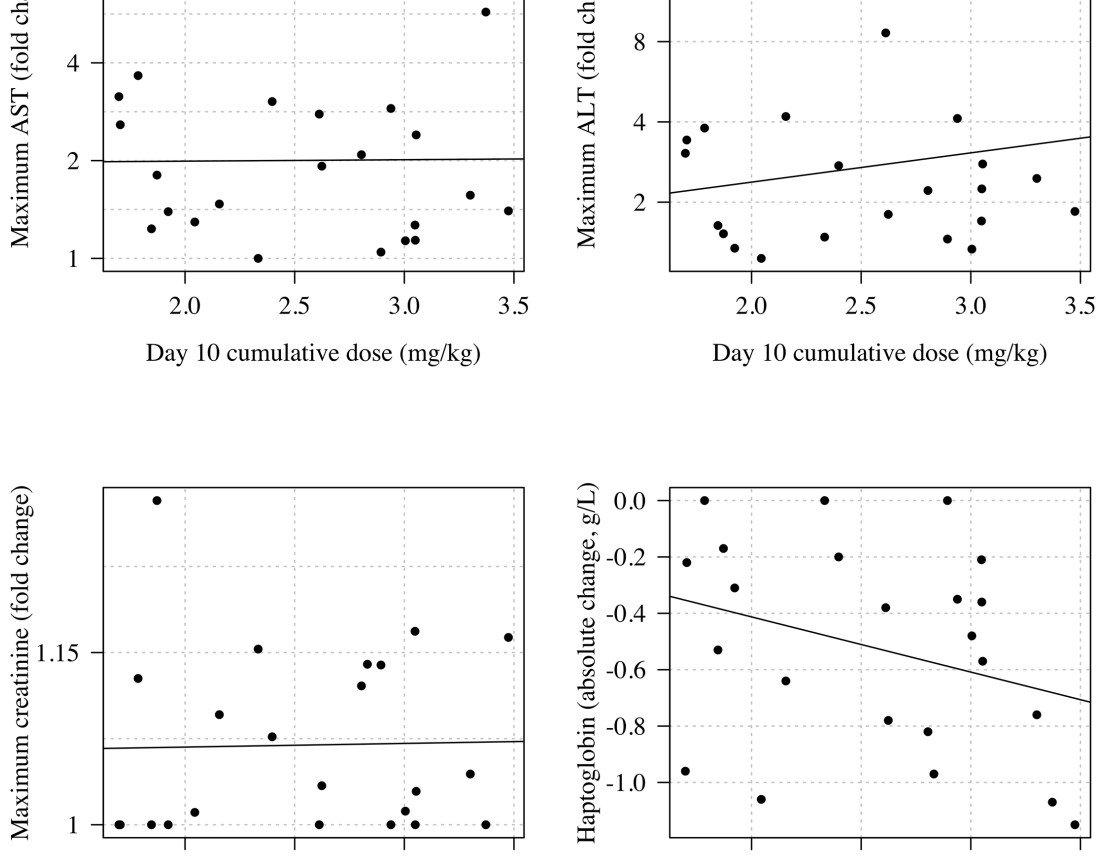

**Appendix 8—figure 1.** Normalised changes in the plasma concentrations of transaminases, haptoglobin, and creatinine during the ascending dose regimen. Colours correspond to the day 10 cumulative dose (as in *Figure 1* in main text). The numbers in the top panels highlight the data from subjects 7, 11, and 14 who had substantial raises in AST and ALT.

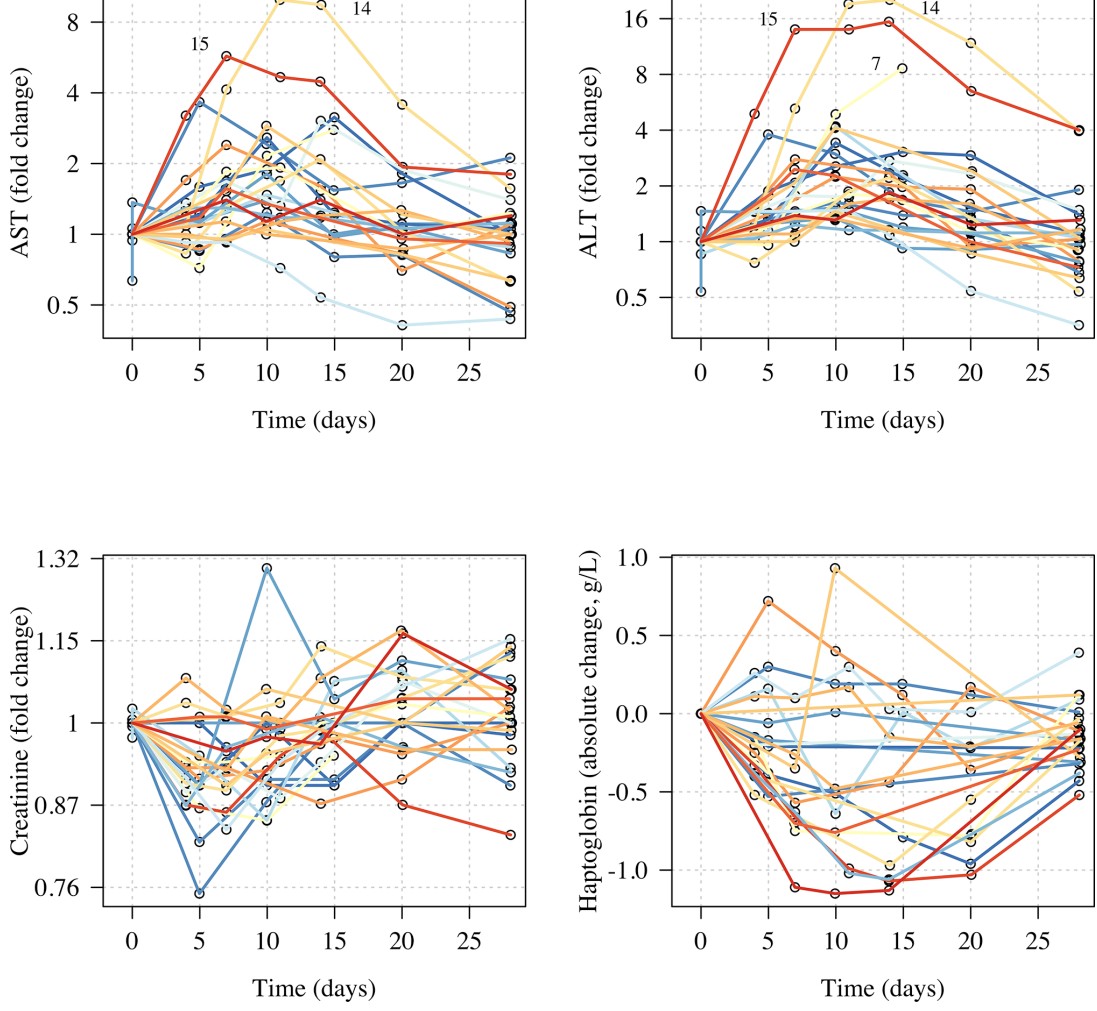

**Appendix 8—figure 2.** Normalised changes in the plasma concentrations of transaminases, haptoglobin, and creatinine during the single dose primaquine 45 mg study.

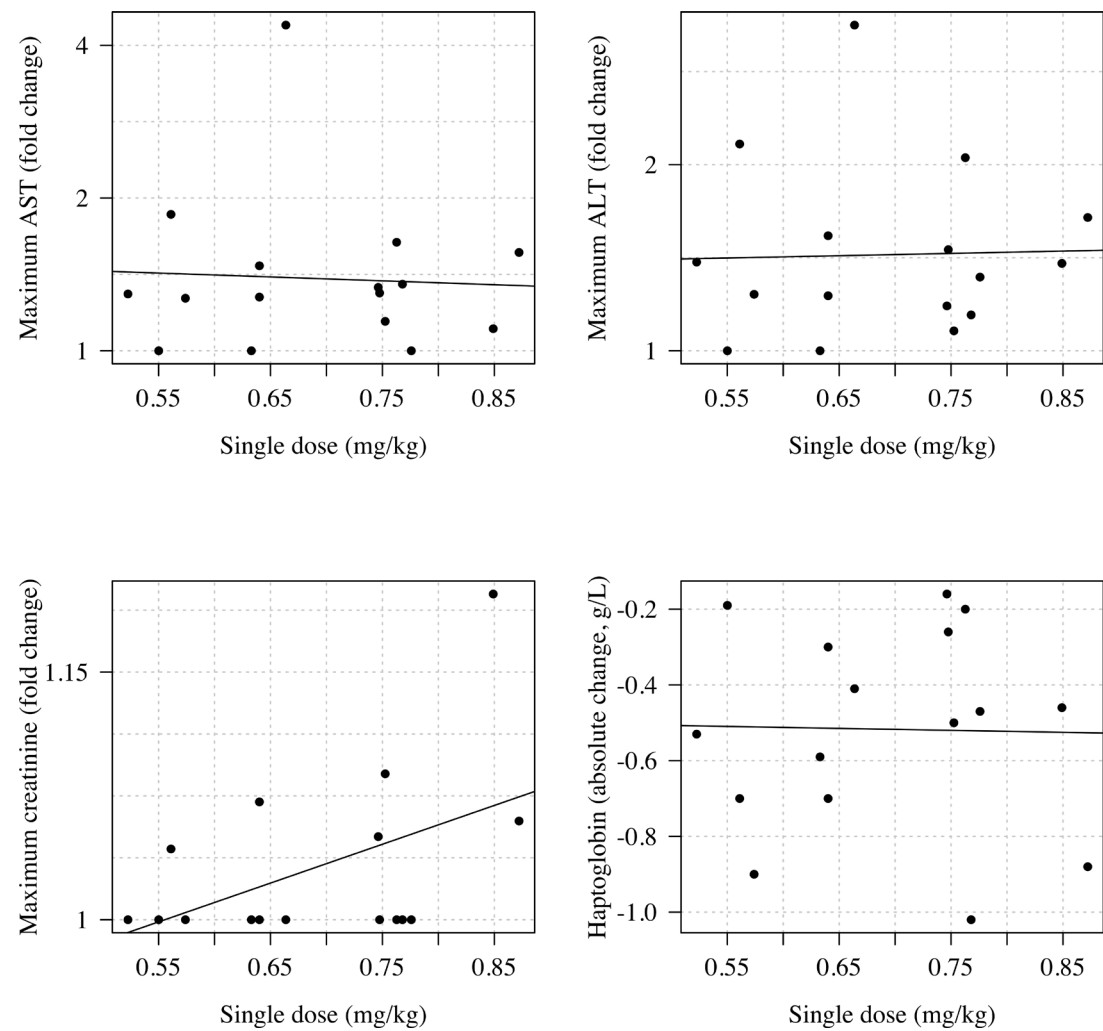

**Appendix 8—figure 3.** Relationship between the total cumulative dose of primaquine received by day 10 in the 23 volunteers who took ascending primaquine dose regimens and the maximum fold change increases in AST, ALT, and creatinine and the maximum absolute decrease in haptoglobin.

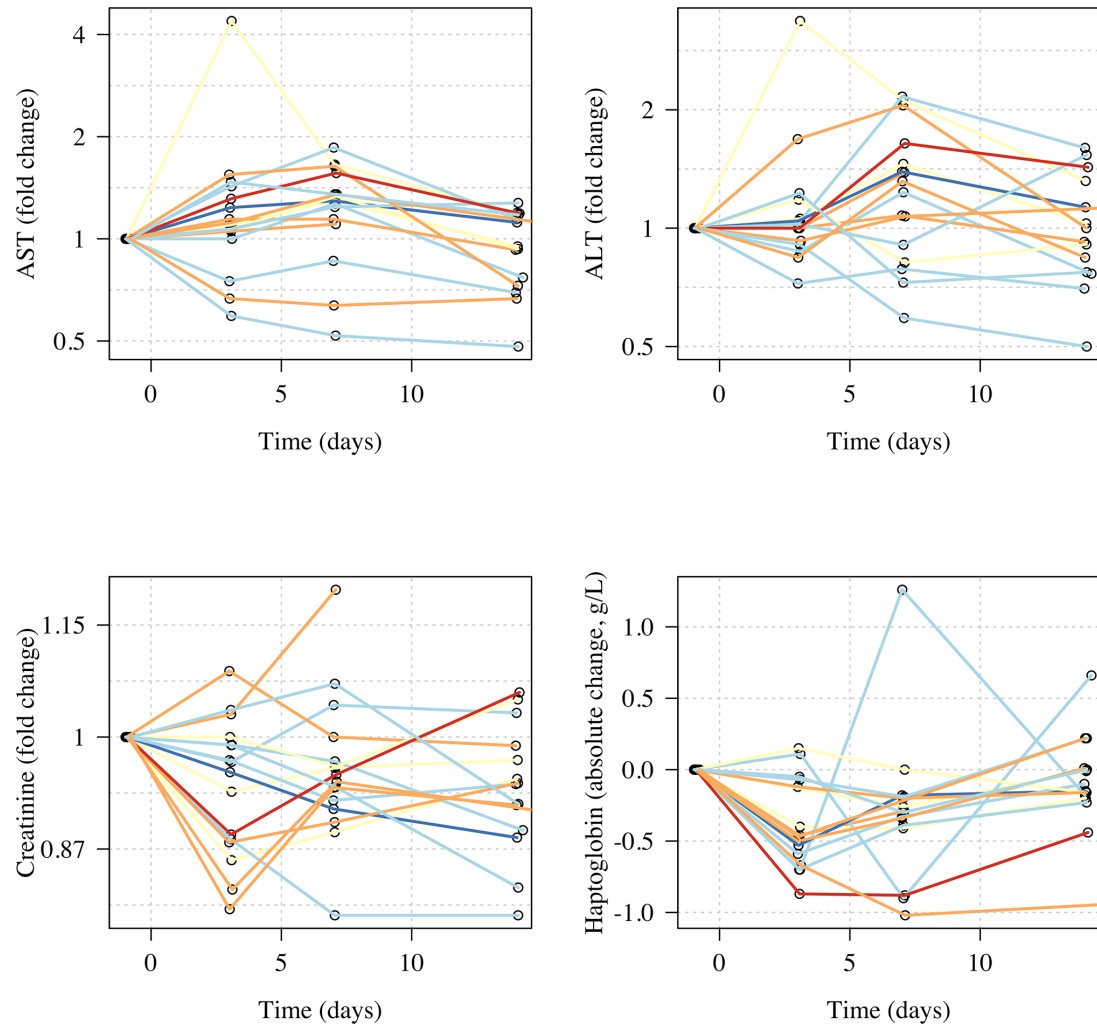

**Appendix 8—figure 4.** Relationship between the mg/kg single dose of primaquine received by the 16 volunteers and the maximum fold change increases in AST, ALT, and creatinine and the maximum absolute decrease in haptoglobin.

## Appendix 9

### Part 1: Individual patient data

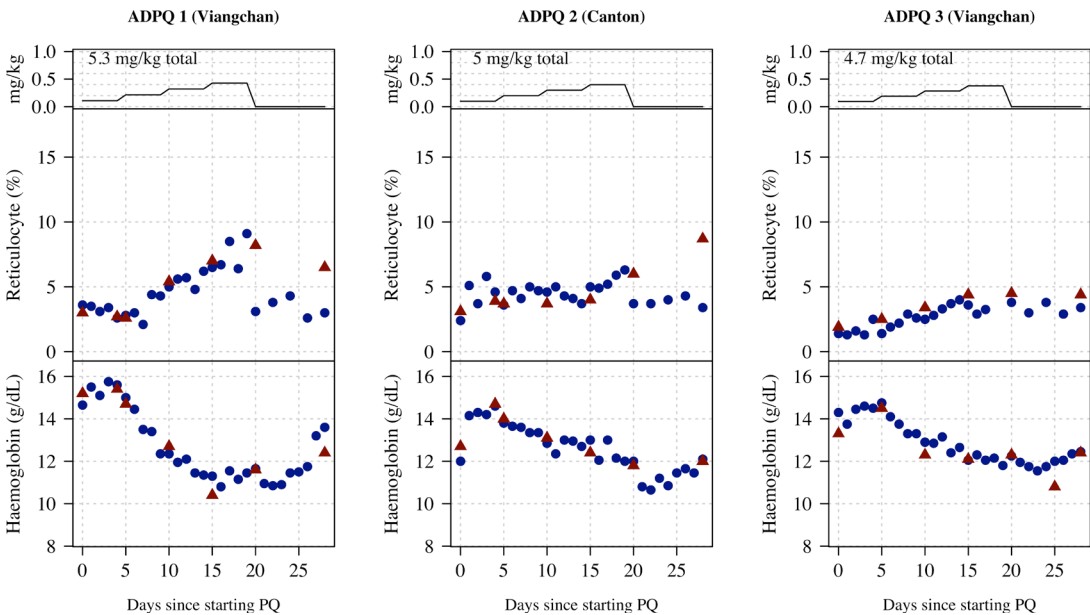

**Appendix 9—figure 1.** Ascending dose: haemoglobin and reticulocyte data over time for volunteers 1–3. Blue shows the manual reticulocyte readings and haemocue value; red shows the complete blood count (CBC) values.

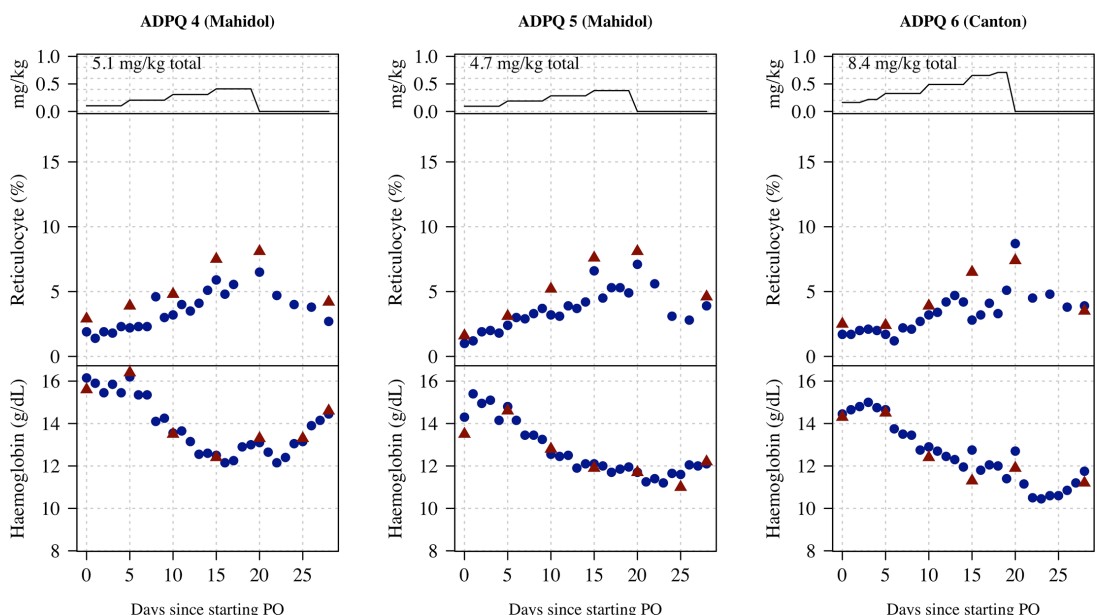

**Appendix 9—figure 2.** Ascending dose: haemoglobin and reticulocyte data over time for volunteers 4–6. Blue shows the manual reticulocyte readings and haemocue value; red shows the complete blood count (CBC) values.

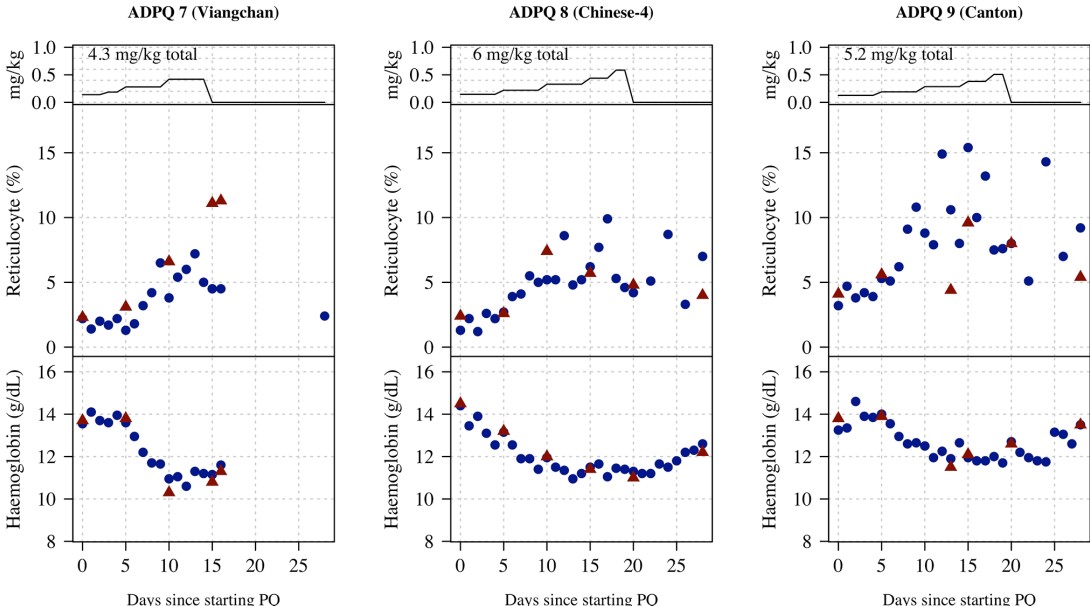

**Appendix 9—figure 3.** Ascending dose: haemoglobin and reticulocyte data over time for volunteers 7–9. Blue shows the manual reticulocyte readings and haemocue value; red shows the complete blood count (CBC) values.

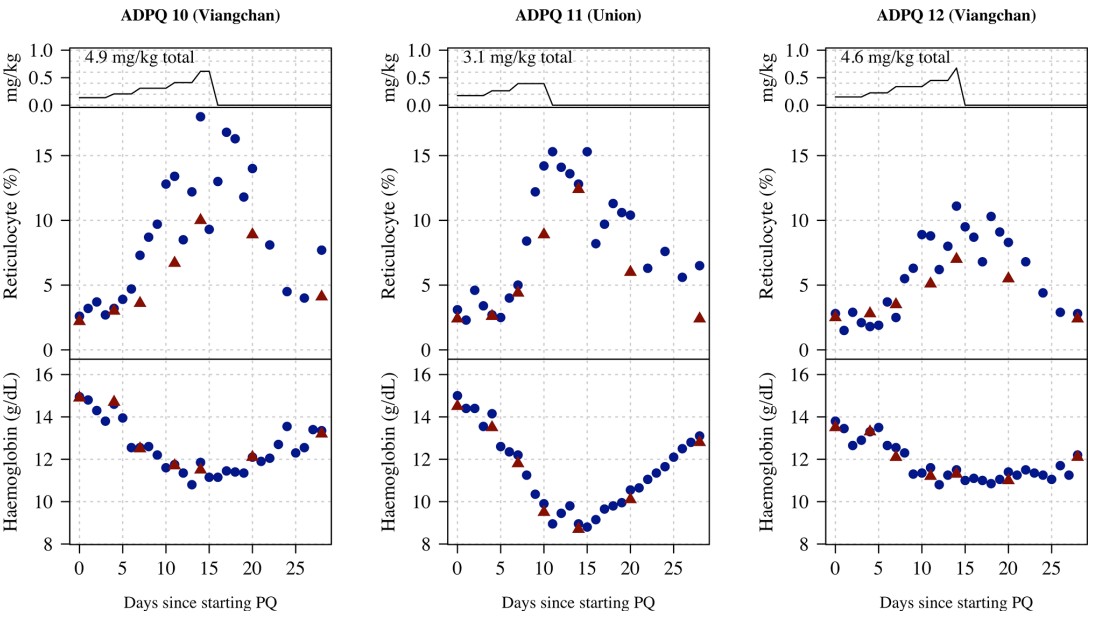

**Appendix 9—figure 4.** Ascending dose: haemoglobin and reticulocyte data over time for volunteers 10–12. Blue shows the manual reticulocyte readings and haemocue value; red shows the complete blood count (CBC) values.

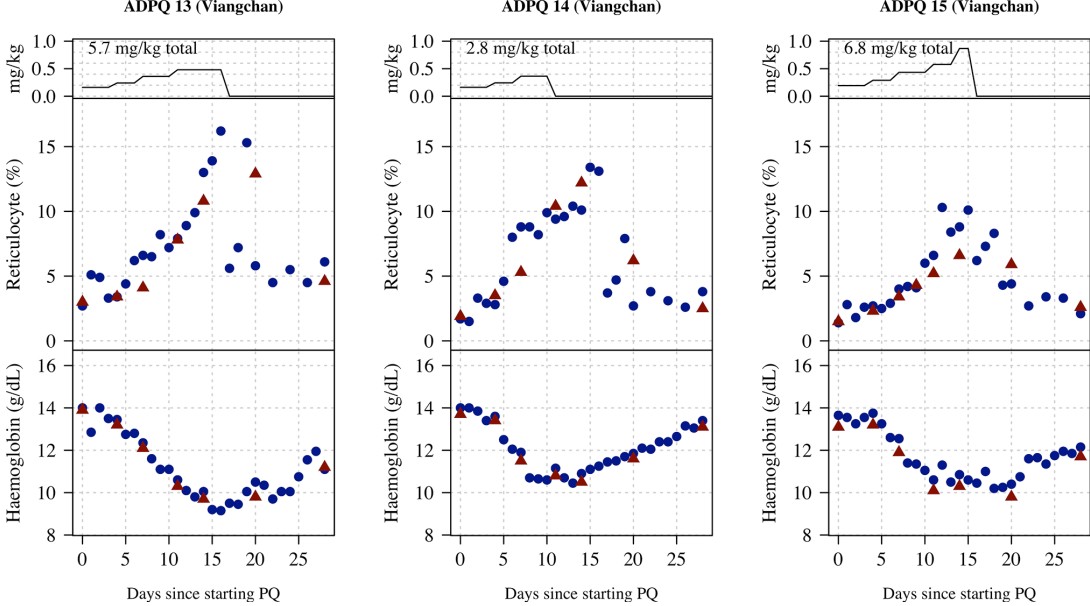

**Appendix 9—figure 5.** Ascending dose: haemoglobin and reticulocyte data over time for volunteers 13–15. Blue shows the manual reticulocyte readings and haemocue value; red shows the complete blood count (CBC) values.

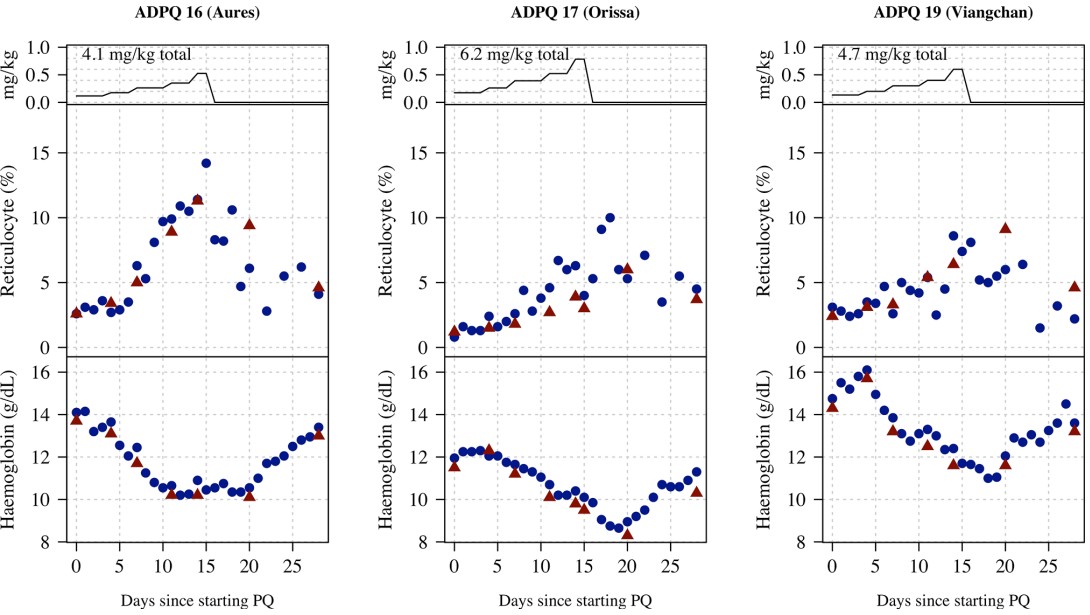

**Appendix 9—figure 6.** Ascending dose: haemoglobin and reticulocyte data over time for volunteers 16–17, 19. Blue shows the manual reticulocyte readings and haemocue value; red shows the complete blood count (CBC) values.

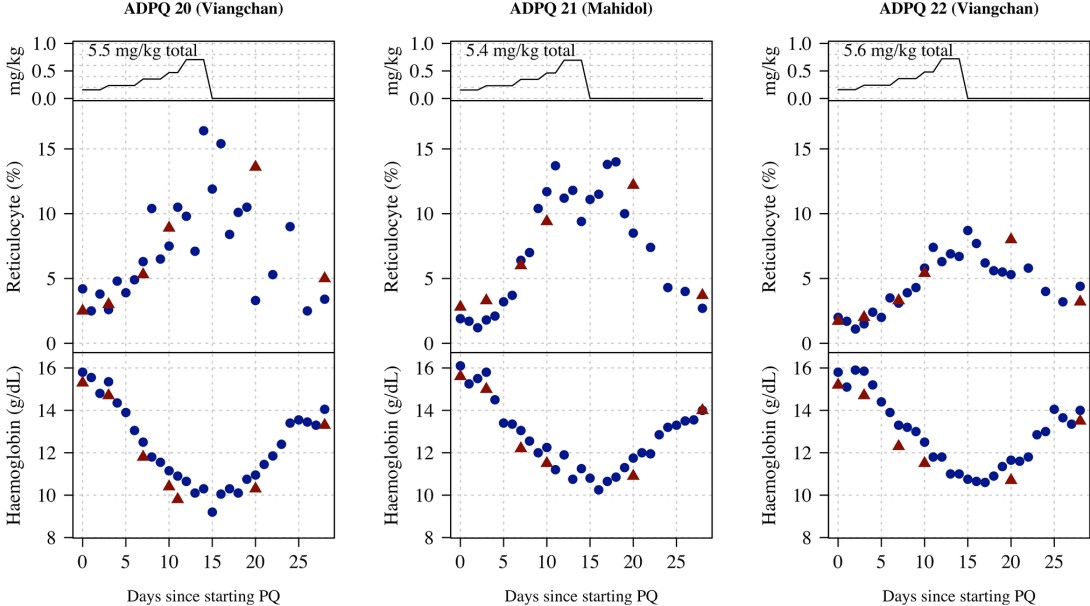

**Appendix 9—figure 7.** Ascending dose: haemoglobin and reticulocyte data over time for volunteers 20–22. Blue shows the manual reticulocyte readings and haemocue value; red shows the complete blood count (CBC) values.

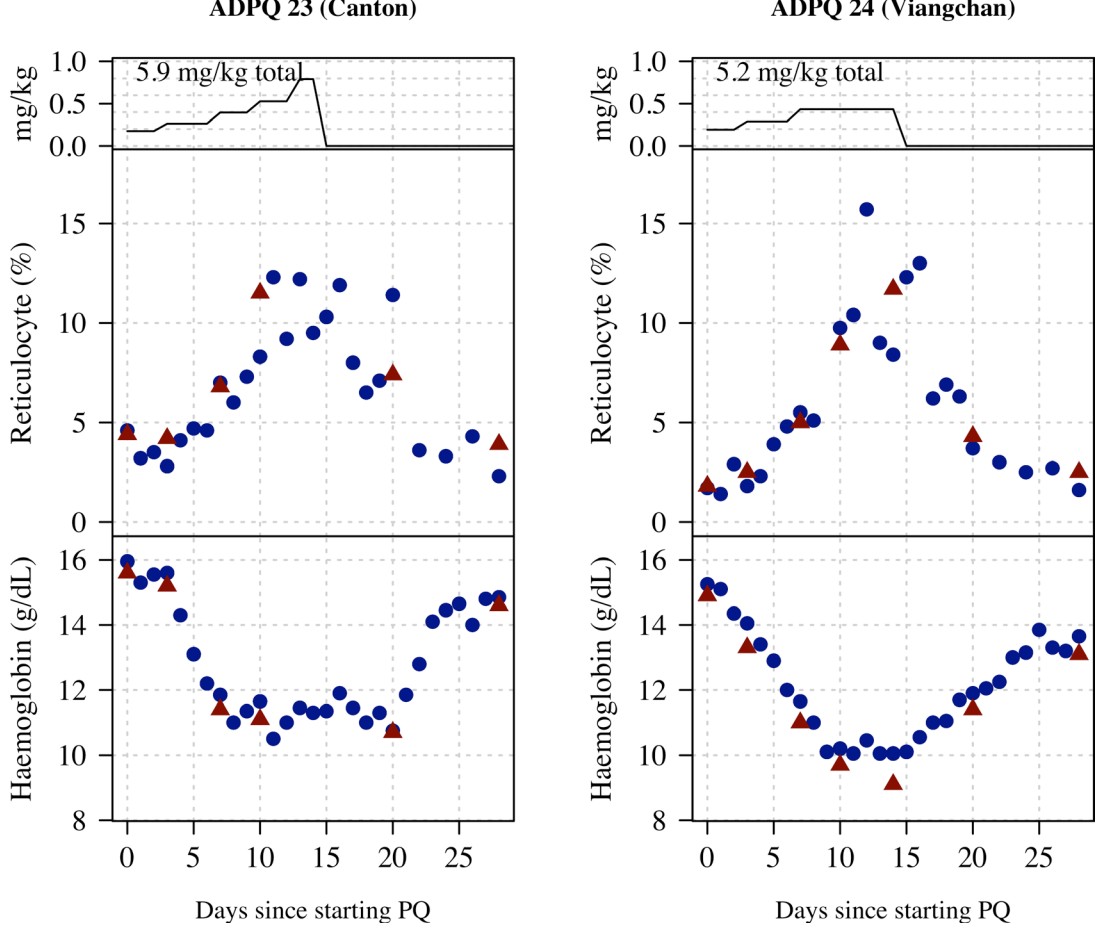

**Appendix 9—figure 8.** Ascending dose: haemoglobin and reticulocyte data over time for volunteers 23 and 24. Blue shows the manual reticulocyte readings and haemocue value; red shows the complete blood count (CBC) values.

## Part 2: Individual patient data

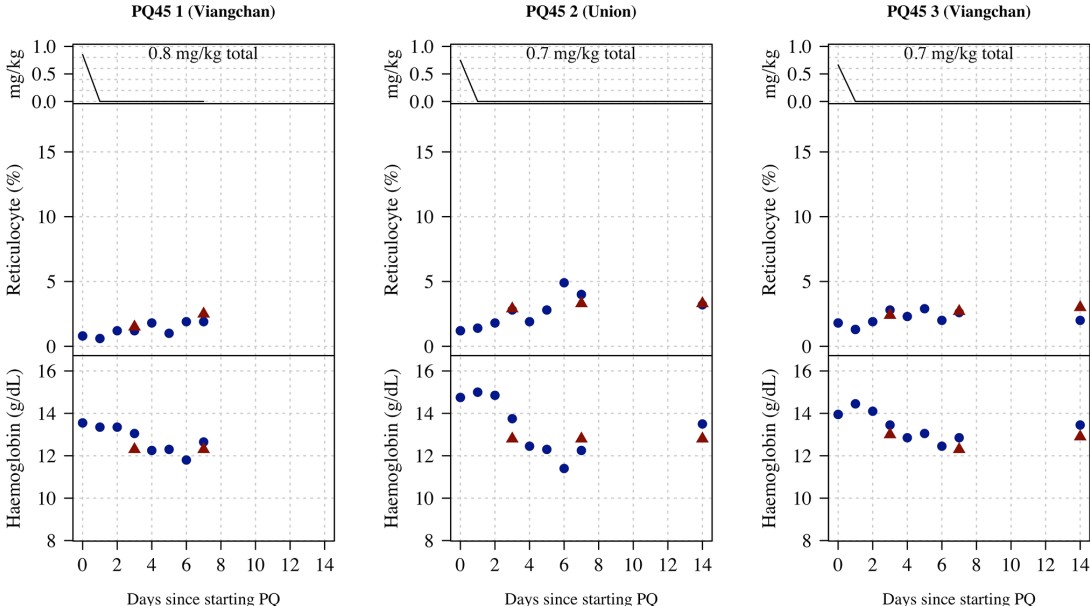

**Appendix 9—figure 9.** Ascending dose: haemoglobin and reticulocyte data over time for volunteers 1–3. Blue shows the manual reticulocyte readings and haemocue value; red shows the complete blood count (CBC) values.

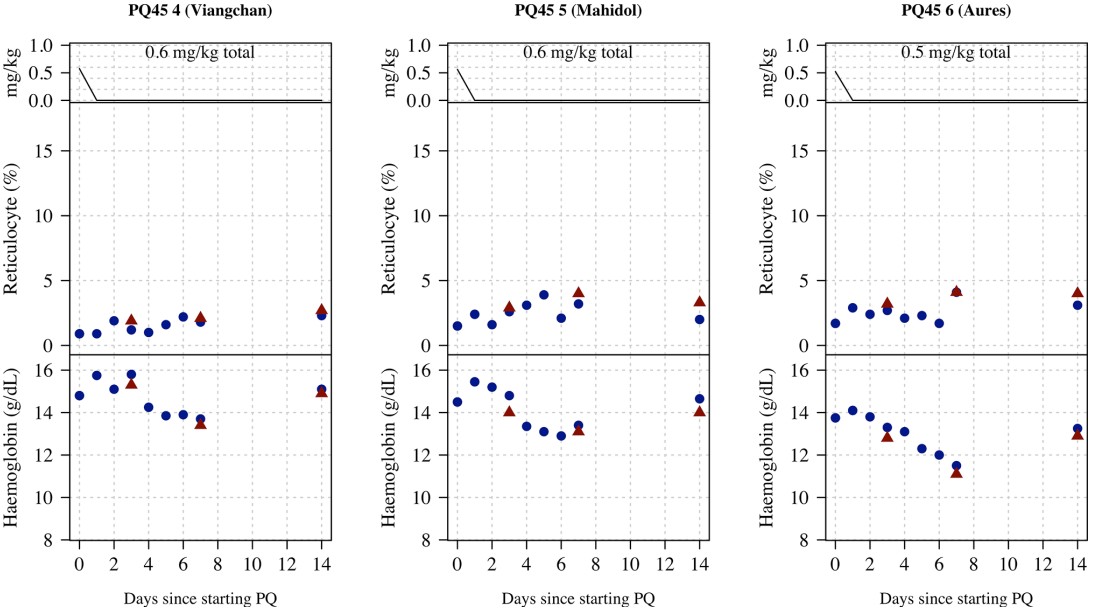

**Appendix 9—figure 10.** Ascending dose: haemoglobin and reticulocyte data over time for volunteers 4–6. Blue shows the manual reticulocyte readings and haemocue value; red shows the complete blood count (CBC) values.

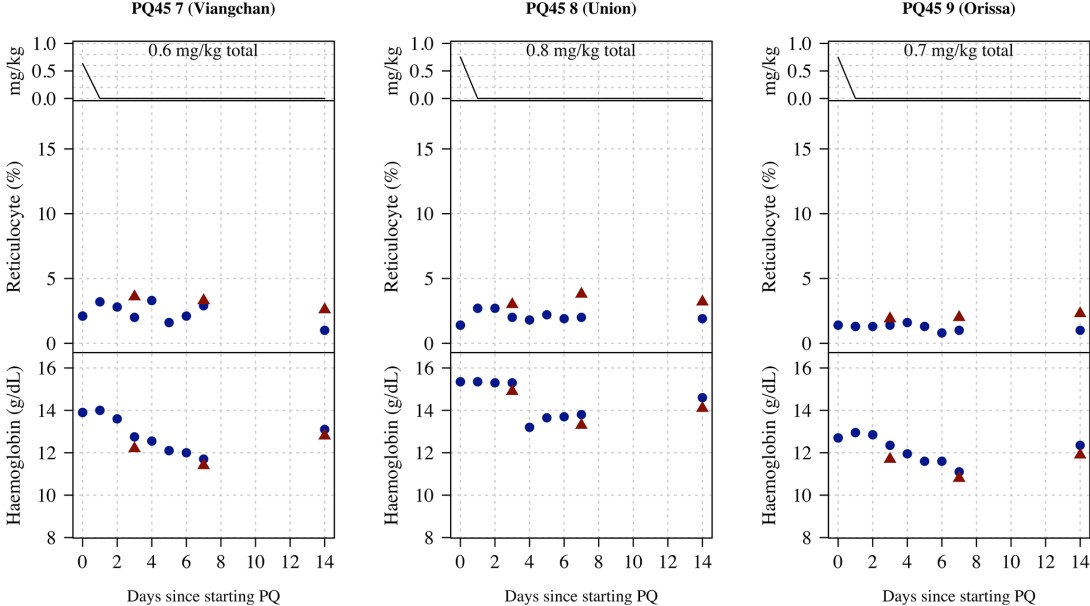

**Appendix 9—figure 11.** Ascending dose: haemoglobin and reticulocyte data over time for volunteers 7–9. Blue shows the manual reticulocyte readings and haemocue value; red shows the complete blood count (CBC) values.

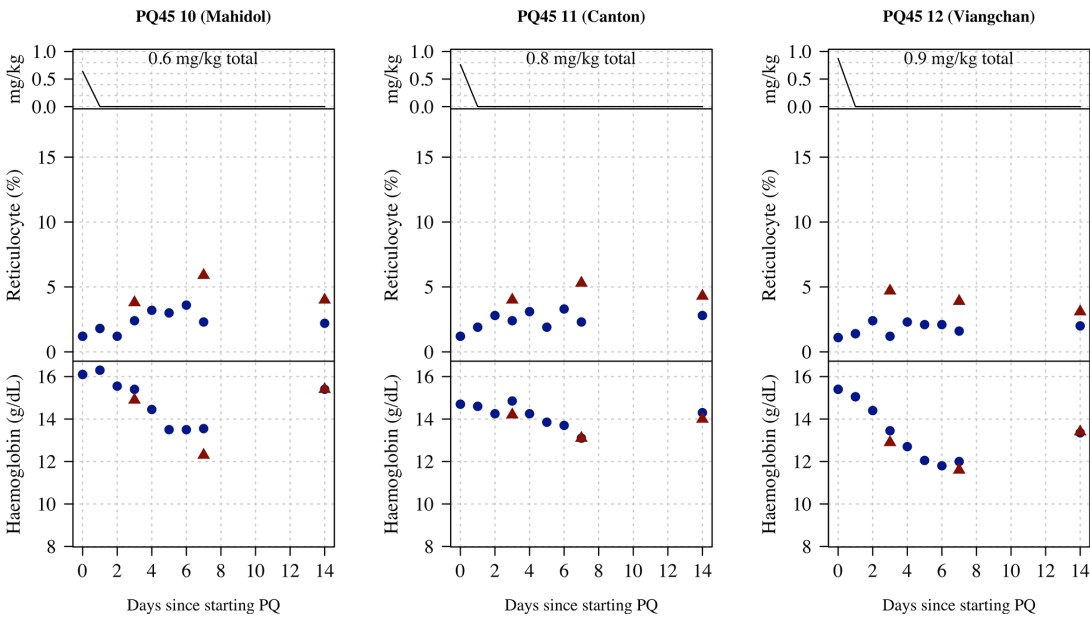

**Appendix 9—figure 12.** Ascending dose: haemoglobin and reticulocyte data over time for volunteers 10–12. Blue shows the manual reticulocyte readings and haemocue value; red shows the complete blood count (CBC) values.

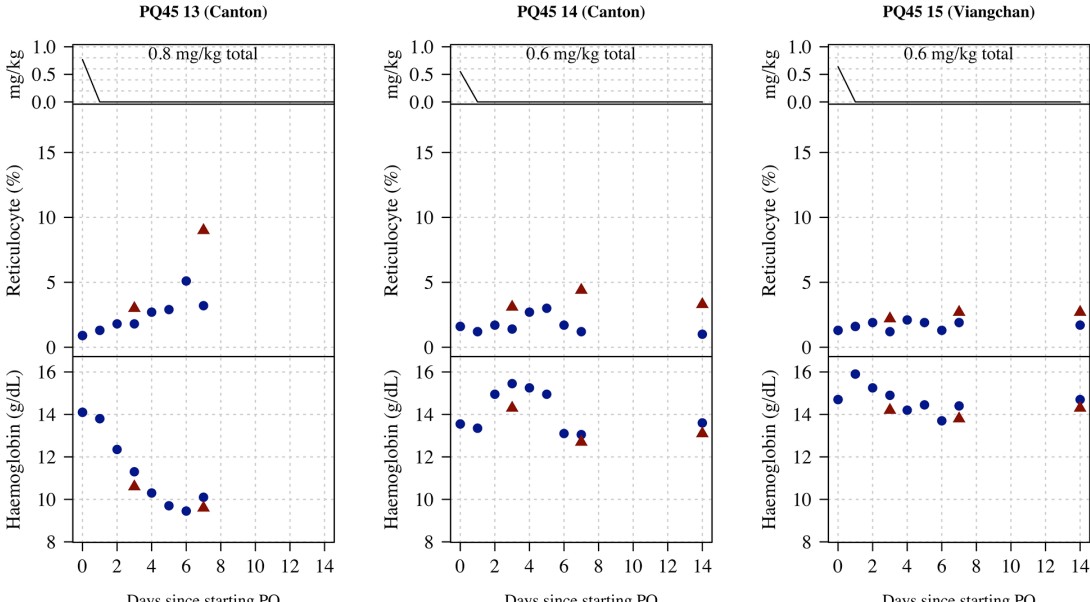

**Appendix 9—figure 13.** Ascending dose: haemoglobin and reticulocyte data over time for volunteers 13–15. Blue shows the manual reticulocyte readings and haemocue value; red shows the complete blood count (CBC) values.

# PQ45 16 (Kaiping)

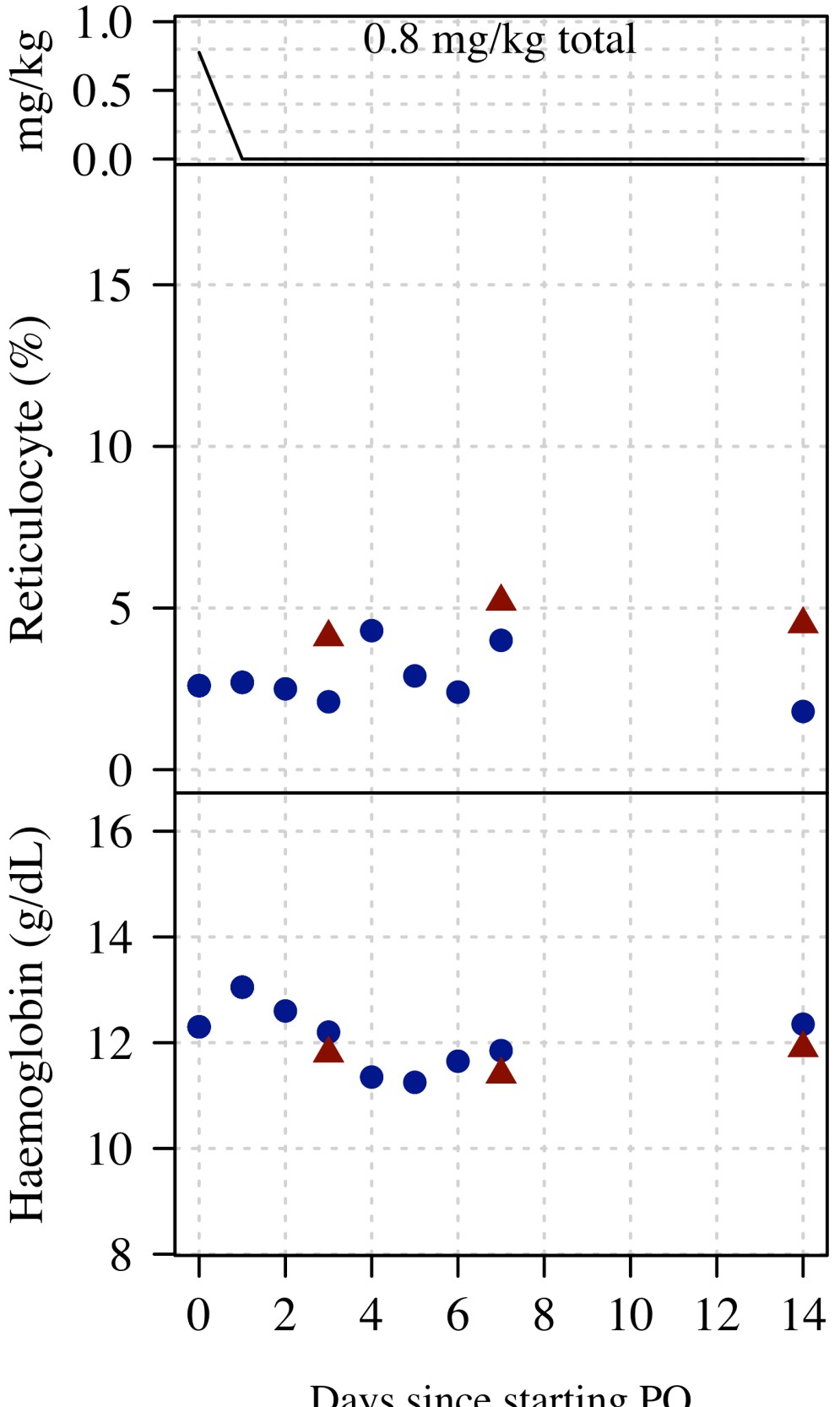

**Appendix 9—figure 14.** Ascending dose: haemoglobin and reticulocyte data over time for volunteer 16. Blue shows the manual reticulocyte readings and haemocue value; red shows the complete blood count (CBC) values.

## Appendix 10

### Comparison of CBC and haemocue

*Figure 1* compares haemoglobin values at timepoints where patients had both CBC and haemocue measured. There is a systematic bias at lower haemoglobin values whereby the CBC gives lower recordings relative to the haemocue. We did not adjust for this in the analysis specifically but report it for interest.

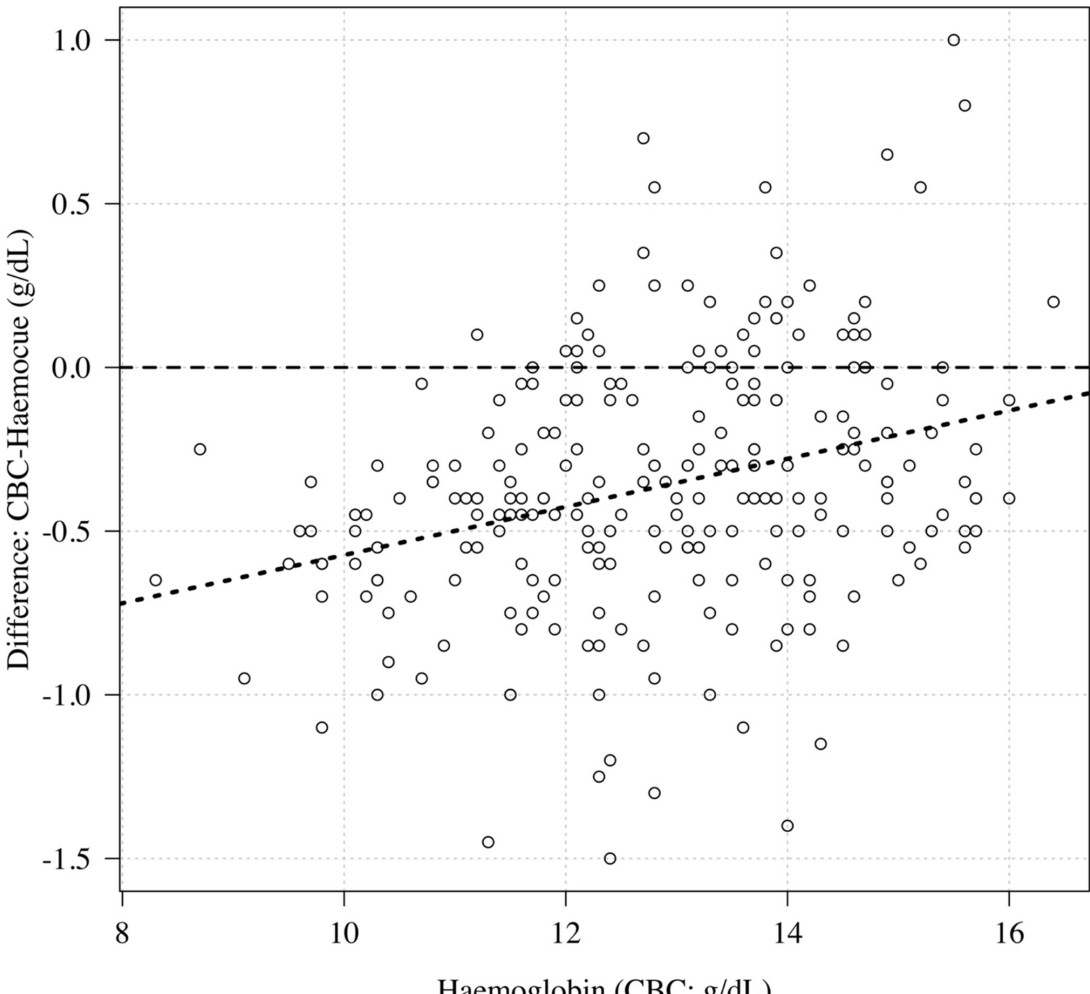

**Appendix 10—figure 1.** Comparison of haemoglobin as measured by complete blood count (CBC) (x-axis) and haemocue (difference between CBC and haemocue shown on the y-axis). The haemocue gave systematically higher haemoglobin concentrations relative to the CBC for low haemoglobin concentrations (about 0.5 g/dL higher at a haemoglobin of 10 g/dL).

