## [Editor Report · eLife assessment]

This manuscript addresses an **important** question, that in countries endemic for *P vivax* the need to administer a primaquine (PQ) course adequate to prevent relapse in G6PD deficient persons poses a real dilemma. On one hand PQ will cause haemolysis; on the other hand, without PQ the chance of relapse is very high. As a result, out of fear of severe haemolysis, PQ has been under-used. This manuscript is **convincing** that regimen (1) can be used successfully to deliver within 3 weeks, under hospital conditions, the dose of PQ required to prevent *P vivax* malaria relapse.

---

## [Referee Report · Joint Public Review]

In countries endemic for P vivax the need to administer a primaquine (PQ) course adequate to prevent relapse in G6PD deficient persons poses a real dilemma. On one hand PQ will cause haemolysis; on the other hand, without PQ the chance of relapse is very high. As a result, out of fear of severe haemolysis, PQ has been under-used.

In view of the above, the authors have investigated in well-informed volunteers, who were kept under close medical supervision in hospital throughout the study, two different schedules of PQ administration: (1) escalating doses (to a total of 5-7 mg/kg); (2) single 45 mg dose (0.75 mg/kg).

It is shown convincingly that regimen (1) can be used successfully to deliver within 3 weeks, under hospital conditions, the dose of PQ required to prevent P vivax relapse.

As expected, with both regimens acute haemolytic anaemia (AHA) developed in all cases. With regimen (2), not surprisingly, the fall in Hb was less, although it was abrupt. With regimen (1) the average fall in Hb was about 4 G. Only in one subject the fall in Hb mandated termination of the study.

Since the data from the Chicago group some sixty years ago, there has been no paper reporting a systematic daily analysis of AHA in so many closely monitored subjects with G6PD deficiency. The individual patient data in the Supplementary material are most informative and more than precious.

Comments on the revised version:

In my view this important paper is further improved in this revised version (R2), particularly with respect to clarity in the discussion. All the points I had previously raised have been tackled.

---

## [Author Response]

The following is the authors’ response to the previous reviews

**Reviewer #1 (Public Review):**
In countries endemic for P vivax the need to administer a primaquine (PQ) course adequate to prevent relapse in G6PD deficient persons poses a real dilemma. On one hand PQ will cause haemolysis; on the other hand, without PQ the chance of relapse is very high. As a result, out of fear of severe haemolysis, PQ has been under-used.In view of the above, the Authors have investigated in well-informed volunteers, who were kept under close medical supervision in hospital throughout the study, two different schedules of PQ administration: (1) escalating doses (to a total of 5-7 mg/kg); (2) single 45 mg dose (0.75 mg/kg).It is shown convincingly that regimen (1) can be used successfully to deliver within 3 weeks, under hospital conditions, the dose of PQ required to prevent P vivax relapse.As expected, with both regimens acute haemolytic anaemia (AHA) developed in all cases. With regimen (2), not surprisingly, the fall in Hb was less, although it was abrupt. With regimen (1) the average fall in Hb was about 4 G. Only in one subject the fall in Hb mandated termination of the study.Since the data from the Chicago group some sixty years ago, there has been no paper reporting a systematic daily analysis of AHA in so many closely monitored subjects with G6PD deficiency. The individual patient data in the Supplementary material are most informative and more than precious.Having said this, I do have some general comments.1. Through their remarkable Part 1 study, the Authors clearly wish to set the stage for a revision of the currently recommended PQ regimen for G6PD deficient patients. They have shown that 5-7 mg/kg can be administered within 3 weeks, whereas the currently recommended regimen provides 6 mg/kg over no less than 8 weeks.

We state in the abstract:“The aim was to explore shorter and safer primaquine radical cure regimens compared to the currently recommended 8-weekly regimen (0.75 mg/kg once weekly), potentially obviating the need for G6PD testing”. This is the primary goal of the study.

1. Part 2 aims to show that, as was known already, even a single PQ dose of 0.75 mg/kg causes a significant degree of haemolysis: G6PD deficiency-related haemolysis is characteristically markedly dose-dependent. Although they do not state it explicitly in these words (I think they should), the Authors want to make it clear that the currently recommended regimen does cause AHA.

We also wanted to compare the extent of haemolysis following single dose with the extent of haemolysis following the ascending dose regimens, in the same patients.

1. Regulatory agencies like to classify a drug regimen as either SAFE or NOT-SAFE; they also like to decide who is 'at risk' and who is 'not at risk'. A wealth of data, including those in this manuscript, show that it is not correct to say that a G6PD deficient person when taking PQ is at risk of haemolysis: he or she will definitely have haemolysis. As for SAFETY, it will depend on the clinical situation when PQ is started and on the severity of the AHA that will develop.

We agree completely. Haemolysis following primaquine is inevitable. What matters is the rate and extent of haemolysis, and the compensatory response. Importantly the extent of the haemolysis, even within a specific genotype and for a given drug dose, appears to be highly variable.

The above three issues are all present in the discussion, but I think they ought to be stated more clearly.

We have tried to clarify these points in a revised discussion.

Finally, by the Authors' own statement on page 15, the main limitation is the complexity of this approach. The authors suggest that blister packed PQ may help; but to me the real complexity is managing patients in the field versus the painstaking hospital care in the hands of experts, of which volunteers in this study have had the benefit. It is not surprising that a fall in Hb of 4 g/dl is well tolerated by most non-anaemic men; but patients with P vivax in the field may often have mild to moderate to severe anaemia; and certainly they will not have their Hb, retics and bilirubin checked every day. In crude approximation, we are talking of a fall in Hb of 4 G with regimen (1), as against a fall in Hb of 2 G with regimen (2), that is part of the currently recommended regimen: it stands to reason that, in terms of safety, the latter is generally preferable (even though some degree of fall in Hb will recur with each weekly dose). In my view, these difficult points should be discussed deliberately.

As above we have tried to clarify these important points in a revised discussion

**Reviewer #1 (Recommendations For The Authors):**
Page 2 para 3. The decreased haemolysis upon continued PQ administration (that originally was named the 'resistance phase' is explained by two additive factors. First, the reticulocytosis cells with higher G6PD activity pour into circulation from the bone marrow); second, the early doses of PQ has caused selective haemolysis of the oldest red cells, that had the lowest G6PD activity. This dual phenomenon is hinted at, but I think it should be stated clearly.

Thank you. We have added to the Introduction (fourth paragraph in revised version):

“Continued primaquine administration to G6PD deficient subjects resulted in "resistance" to the haemolytic effect. The selective haemolysis of the older red cells resulted in a compensatory increase in the number of reticulocytes. Thus, the red cell population became progressively younger and increasing resistant to oxidant stress, so overall haemolysis decreased and a steady state was reached.”

Page 4 and elsewhere. In the 'Hillmen scale' for haemoglobinuria a value >6 was named a 'paroxysm'; but any value of 2 and above is already frank haemoglobinuria. Incidentally, the chart was published not in ref 17, but in NEJM 350:552, 2004.

We have changed the reference (now ref 19) to the 2004 paper by Hillmen.We used the value of 6 as clinical criterion for stopping primaquine. While >2 is detectable in dilute urine, >6 refers to clearly red/black urine.

In Table 1 and throughout the paper I am surprised that retics are given as %: absolute retic counts are more informative.

We showed these as % counts as the majority of measurements were taken from blood slide readings where it is not possible to get an absolute count.

Page 10, Attenuated hemolysis with continued or recurrent doses of PQ was shown convincingly for G6PD A-. There is also one report in which the time course of AHA was extensively investigated upon deliberate administration of PQ to a subject with G6PD Mediterranean (Blood 25: 92, 1965): there was little or no evidence for a 'resistance phase'.

We agree that this suggests it might not be possible to attenuate haemolysis with the Mediterranean variant (or variants of similar severity) as even the youngest circulating red cells may be susceptible to haemolysis. More evidence is needed.

S6, S7. Reticulocytes remain high until PQ is stopped; they return to normal some 17 days after stopping PQ. This should be stated in the main text.

This has been added to the main text (section “Haemolysis and reticulocyte response”):

“It took around 2 weeks for the reticulocyte counts to re-normalise.”

In subject 11 haemoglobinuria was slight on day 12; what was it before?

We have changed the caption of this Figure (Appendix 5) to:

“Day 10 urine sample from subject 11 showing slight haemoglobinuria (Hillmen score of 4). The subject had a maximum Hillmen score varying between 2 and 3 on days 4 to 9.”

I found individual patient data in S5 and S6 most interesting, especially since the G6PD variant was identified in each case. It would be helpful if in each case the total PQ dose were also shown, and in the interest of visual comparability the abscissa scale ought to be the same for all cases.

We have amended Figures S5 and S6 to make them consistent with each other (now Appendix 5). We also amended the figures showing the individual subject data for consistency.